# Engineering tumoral vascular leakiness with gold nanoparticles

Magdiel Inggrid Setyawati [1,7,8] ✉, Qin Wang [2,8], Nengyi Ni[1], Jie Kai Tee [3], Katsuhiko Ariga [4,5], Pu Chun Ke [6], Han Kiat Ho [3], Yucai Wang [2] ✉ & David Tai Leong [1] ✉

Delivering cancer therapeutics to tumors necessitates their escape from the surrounding blood vessels. Tumor vasculatures are not always sufficiently leaky. Herein, we engineer therapeutically competent leakage of therapeutics from tumor vasculature with gold nanoparticles capable of inducing endothelial leakiness (NanoEL). These NanoEL gold nanoparticles activated the loss of endothelial adherens junctions without any perceivable toxicity to the endothelial cells. Microscopically, through real time live animal intravital imaging, we show that NanoEL particles induced leakiness in the tumor vessels walls and improved infiltration into the interstitial space within the tumor. In both primary tumor and secondary micrometastases animal models, we show that pretreatment of tumor vasculature with NanoEL particles before therapeutics administration could completely regress the cancer. Engineering tumoral vasculature leakiness represents a new paradigm in our approach towards increasing tumoral accessibility of anti-cancer therapeutics instead of further increasing their anti-cancer lethality.

One of the most fundamental assumptions of all intravenous cancer therapeutics is that they must cross the endothelial barrier to exert their therapeutic effects. The ability to control this endothelial accessibility not only determine the therapeutic outcome but also any detrimental off-target effects. The leakiness of the cancer vasculature is largely believed to be ironically dependent on the tumor itself. Nanomedicine depends on this tumor-dependent endothelial leakiness, endothelial permeability and retention (EPR) effect to escape from vasculature and access the tumor. The EPR effect came about due to metabolic deficiencies in tumors of certain maturity and size that allow them to secrete pro-angiogenic factors to secure their own blood

supply from the surrounding otherwise healthy vasculature. The tumor vasculature tends to be randomly leaky and uncontrollable in degree and time extent. Late-stage tumor cells also exploit their self-induced endothelial leakiness to metastasize. Normalization of these cancer vasculatures does stem the nutrient spike for the tumor but may also reduce therapeutics access to the tumor. So an in-between strategy might be to engineer, at will, temporary endothelial leakiness.

We have shown earlier that certain nanoparticles, independent of cancer cells, are able to induce in vitro and in vivo endothelial leakiness (nanoparticle-induced endothelial leakiness or NanoEL)[1,2] in the form of intercellular gaps that ranges from nanometers to a few

[1]National University of Singapore, Department of Chemical and Biomolecular Engineering, 4 Engineering Drive 4, Singapore 117585, Singapore. [2]Department of Radiology, The First Affiliated Hospital of University of Science and Technology of China, Division of Life Sciences and Medicine, University of Science and Technology of China, Hefei 230001, China. [3]National University of Singapore, Department of Pharmacy, 18 Science Drive 4, Singapore 117543, Singapore. [4]WPI Research Center for Materials Nanoarchitectonics (MANA), National Institute for Materials Science (NIMS), 1-1 Namiki, Tsukuba, Ibaraki 305-0044, Japan. [5]Graduate School of Frontier Sciences, The University of Tokyo, 5-1-5 Kashiwanoha, Kashiwa, Chiba 277-8561, Japan. [6]ARC Centre of Excellence in Convergent Bio-Nano Science and Technology, Monash Institute of Pharmaceutical Sciences, Monash University, 381 Royal Parade Parkville, Melbourne, VIC 3052, Australia. [7]Present address: Nanyang Technological University, School of Materials Science and Engineering, 50 Nanyang Avenue, Singapore 639798, Singapore. [8]These authors contributed equally: Magdiel Inggrid Setyawati, Qin Wang. ✉e-mail: misetyawati@ntu.edu.sg; yucaiwang@ustc.edu.cn; cheltwd@nus.edu.sg

micrometers in diameter[1,2]. Previously, we have systematically showed that in vitro NanoEL induction is governed by the NPs' intrinsic particle density[3], size[4,5], and surface charge[6]. We therefore proposed to use a non-tumor dependent NanoEL concept to induce endothelial leakiness at the tumor vasculature and open up that therapeutic time window of opportunity to access the tumor for nanomedicine and drugs; especially in situations where there is little EPR effect. The gaps can provide an uninhibited path that follows the pressure gradient from the blood vessels to the interstitial space of the tumor, thereby circumventing the need for any endothelial energetically costly strategies like transcytosis[7,8]. Moreover, sole reliance on tumor-induced endothelial leakiness, commonly known as the endothelial permeability and retention (EPR) effect does not appear to be therapeutically optimistic[9] and may not be present in all tumors. Since the tumor is the sole determinant of the exploitable EPR effect, nanomedicine is at the mercy of the tumors. Because NanoEL does not require cellular uptake of any nanoparticles by the endothelium, the endothelium will not be collaterally damaged. Recovery from NanoEL-induced intercellular gaps are intrinsically hotwired into the endothelial biology as seen in the physiological vascular extravasation of leukocytes; after traversing across the endothelial barrier; it promptly re-establishes[10,11].

In this study, using a library of gold nanoparticles of different sizes and degrees of roughness, we created a calibratable leakiness effect in vivo. Here, we show with intravital imaging of in vivo cancer models that we could induce endothelial leakiness in primary tumors that have little perceivable vasculature leakiness. Utilizing NanoEL gold particles, engineered endothelial leakiness leads to increased leakage of even larger-sized nanoparticles from the tumor vasculature flow to the interstitial space. We also show that the increased leakage could bring about complete regression of the primary tumors without increasing metastasis in some cases. Considering that metastasized tumors account for most cancer deaths, we also applied NanoEL on recently metastasized secondary tumors called micrometastases. Here, we also successfully attacks the secondary metastatic tumor when it is at the micrometastases stage by inducing NanoEL and then treating the micrometastases. In vivo NanoEL effect may have an amplifying action on non-leaky but weakened tumor vessel walls as in the case of pathological neovascularization from tumors to bring about robust penetration of therapeutics into the tumor. In vivo NanoEL does not appear to be affecting healthy vessels. NanoEL presents an engineerable strategy to increase therapeutics access to tumors and perhaps applicable to other diseases that are intimately linked to the vasculature.

## Results and discussions

### Characterization of NanoEL Au NPs library

We synthesized a library of Au NPs comprised of two size series ($Au_{30}$ and $Au_{70}$ series) with four different surface roughness groups (smooth, $R_0$; low roughness $R_1$; mid roughness $R_2$; and high roughness, $R_3$), as depicted in the transmission electron microscope (TEM) images in Fig. 1a and Supplementary Fig. 1a. As Au NPs possess a unique optical property, we were able to detect the change in surface roughness with the UV-Vis spectroscopy analysis[12]. In the $Au_x$ series, the plasmonic absorption peaks and red shifts showed surface roughness increases (Supplementary Fig. 1b). We also characterized our library in terms of hydrodynamic sizes, zeta potential, and primary sizes (Supplementary Table 1). Our analysis showed that hydrodynamic size (based on number distribution) of these Au NPs did not significantly differ from the primary size obtained from the TEM analysis (Supplementary Table 1). In addition, we did not observe significant differences in terms of hydrodynamic sizes in different roughness group NPs. An increase in the measured hydrodynamic size was detected following the NPs exposure to a complete cell culture medium with high protein content (Supplementary Table 1). We attributed this hydrodynamic size increase to the formation of protein corona on the NPs[13]. As previously

reported, the presence of protein in the cell culture medium could be readily absorbed on to NPs through various interactions, namely the electrostatic and the van der Waals interactions[8,13,14]. The formation of the protein corona could also be detected through the change in the measured $\zeta$–potential. Owing to the presence of the phosphine groups originating from the HEPES capping[12,15], the Au NPs in ultrapure water were measured to bear a negative charge with average registered $\zeta$–potential of −30 mV. The measured $\zeta$–potential value shifted to −16 mV after being exposed to cell culture medium, confirming the protein corona formation on the Au NPs surface (Supplementary Table 1).

### Au NP triggered NanoEL on monolayer of endothelial cells in a size and surface roughness dependent manner

Exposing our library of NanoEL Au particles to a monolayer of endothelial cells, we were able to observe endothelial leakiness gaps of approximately 5-20 μm ranges (Fig. 1b and Supplementary Fig. 2) with more numerous and wider NanoEL gaps being observed on the smaller sized particles with increased surface roughness groups (Fig. 1c, d). This trend is further validated in transwell assays (Fig. 1e). Size influences on NanoEL were similar to previous studies[1,4,5]. Onset of NanoEL by $Au_{30}R_3$ treated group occurred 15 min after exposure compared to 30 min in the smooth $Au_{30}$ NPs ($Au_{30}R_0$) group (Supplementary Fig. 3a), and was further validated in the transwell assays (Fig. S4). The same trend was observed with $Au_{70}$ NPs groups (Supplementary Fig. 3b and Supplementary Fig. 4). When the Au NPs were able to reach the adherens junctions, surface roughness would determine the disruptive binding efficacy of the NPs to VE-cadherin and the resultant macroscopic NanoEL effect. Surface roughness has been documented to minimize the repulsive forces between the NPs and biomolecules, resulting in better adhesion between the NPs and the biomolecule[8,16] resulting in a higher NanoEL induction on rough surface Au NPs groups ($R_{1-3}$) as compared to the smooth surface NPs group ($R_0$).

### Au NanoEL induction adhered well to Type I NanoEL mechanism

Through a series of experiments, we determined that $Au_xR_y$ NanoEL is not a result of nanoparticles-induced toxicity on (Supplementary Fig. 5a, b) or induced oxidative stress in the endothelial cells (Supplementary Figs. 5c, 5d, 6a, 6b), or caused by different profile of nanoparticle deposition (Supplementary Tables 2-4) nor even require endocytosis of the particles into the endothelial cells (Supplementary Fig. 7).

Next, we checked whether $Au_xR_y$ NanoEL followed previously characterized cell signaling mechanisms[1]. Homophilically interacted VE-cadherin (VEC) adherens junction proteins, which are typically found along the intercellular gaps of two neighboring endothelial cells, maintain the barrier integrity[17–19]. Previously, we observed that NPs were able to bind to transmembrane VE-cadherin and disrupt the homophilic interaction among these adherens junction protein pairs as Type I NanoEL[1,3,4,20]. Indeed, our TEM analysis of the ultrathin section of the endothelial monolayer treated with $Au_{30}R_3$ shows the $Au_{30}R_3$ NPs to be adjacent to the NanoEL site (Supplementary Fig. 8). This disruption was marked by the activation of VE-cadherin intracellular signaling[18,19]. Adherens junction disassembly activation was reported to be initiated through phosphorylation of $VEC_{Y658}$ and $VEC_{Y731}$[18,19]. Both residues were observed to be significantly phosphorylated when the endothelial cells were exposed to the Au NPs and positively correlated with the roughness of AuNPs (Fig. 2a and Supplementary Fig. 9a). PP1 inhibitor blocks the Src-kinase that phosphorylated these 2 pivotal residues[21]. PP1 inhibitor treatment significantly reduced the level of phosphorylated VE-cadherin at both residues (Fig. 2a and Supplementary Fig. 9a) and likewise significantly reduced NanoEL levels (Fig. 2b). p120 protein binds to the $VEC_{Y658}$ residue to stabilize VE-cadherin-mediated adherens junctions. Phosphorylation of $VEC_{Y658}$ has been reported to promote VE-cadherin endocytosis from the

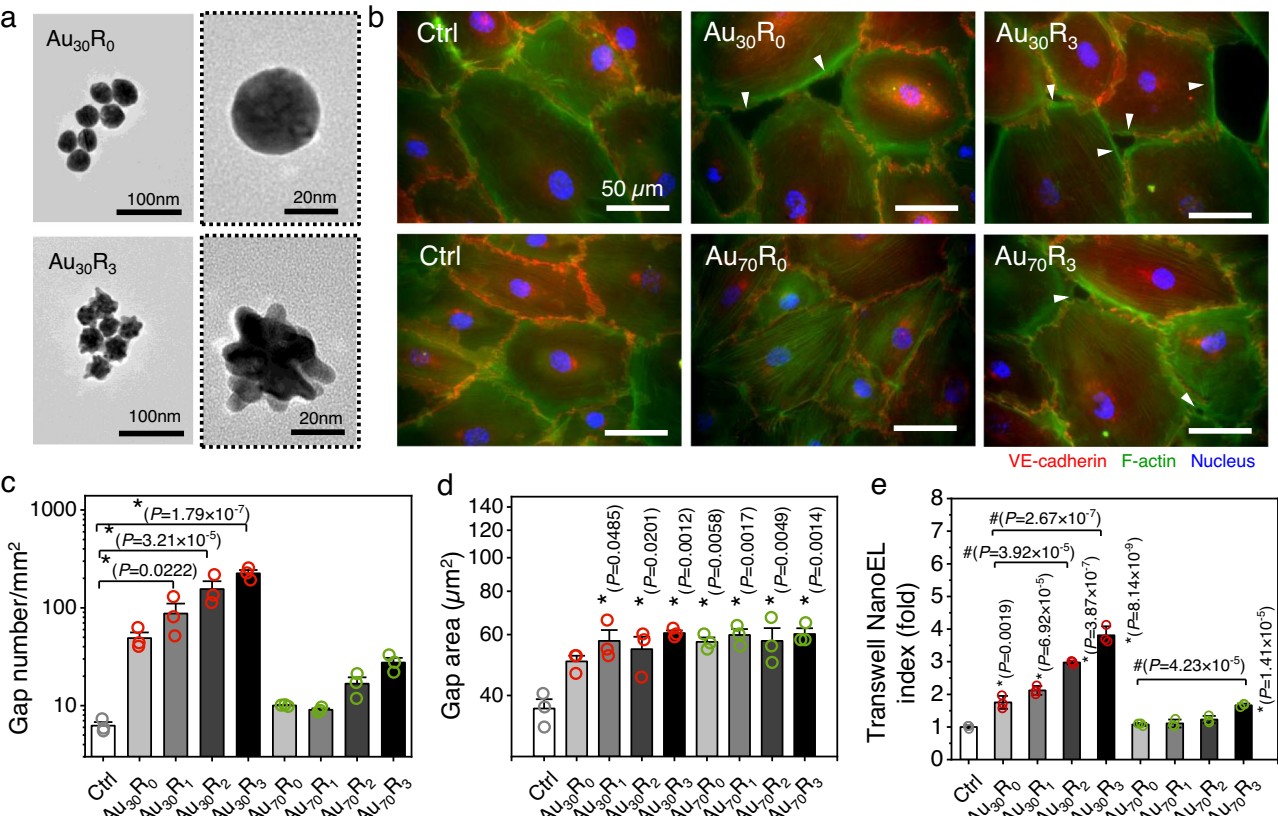

**Fig. 1 | Au NPs induced endothelial cell leakiness (Au NanoEL) in size and surface roughness dependent manner. a** Representative transmission electron microscope (TEM) images depict different surface roughness of Au NPs with primary size 30 nm (Au$_{30}$). High magnification particle image demarcated with dotted line. Images shown are representative of three independent experiments. **b** Immunofluorescence images show the formation of intercellular gaps (white arrowheads) on the monolayer endothelial cell barrier treated with Au NPs (1×10$^9$ particle/mL, 1 h). Scale bar: 50 μm. Nucleus (blue), F-actin (green) and VE-cadherin (red). **c** Number of gaps (n = 12; 4 IF images/group pooled from 3 independent biological replicates/group) and **d** gap size (n = 90; 30 gaps/group pooled from 3

independent biological replicates/group) observed within a certain area. The smaller and rougher the particle, the higher the tendency to induce larger sized gaps and higher gap numbers. The gap area can be used as a gating mechanism for differential access to differently sized entities. **c, d** Data presented in are mean ± SEM. One-way ANOVA, Tukey HSD post-hoc test, *significant against control, P < 0.05. **e** Increase in measured NanoEL index was detected with the increase of Au NPs surface roughness (1×10$^9$ particles/mL, 1 h). Data are mean ± SD, n = 3, One-way ANOVA, Tukey HSD post-hoc test, *significant compared to control, #significant compared to the smooth Au NPs (R$_0$ group), P < 0.05. Source data are provided as a Source Data file.

adherens junction and its proteolytic degradation[18,22]. Treatment of these Au nanoparticles decreased overall VEC expression. Concomitant increased of VEC$_{Y658}$ phosphorylation was observed with the increase of Au NPs roughness. PP1 treatment was also found to significantly rescue the Au NPs reduced VEC expression (Fig. 2a and Supplementary Fig. 9a). VEC$_{Y731}$ residue is a binding site for β-catenin protein, which acts as the bridging protein between the VEC and the actin cytoskeleton[22]. Phosphorylation in this residue was well characterized through the dissociation of VE-cadherin/β-catenin linkage which subsequently activated actin cytoskeleton remodeling. Through our β-catenin immunoprecipitation analysis, we detected significantly decreased amounts of β-catenin associated VE-cadherin protein (Fig. 2c and Supplementary Fig. 9b) with the rougher Au NPs (R$_3$) treated group as compared to the smooth AuNPs (R$_0$) treated group and control. Immunoprecipitation of VE-cadherin showed a reduced level of β-catenin association with VE-cadherin (Fig. 2d and Supplementary Fig. 9c) as expected. The dissociation of VE-cadherin from β-catenin unfetters the actin cytoskeleton[17,18]. This would lead to the remodeling process of the actin skeleton which resulted in the cell contraction and the widening of the intercellular gaps between the endothelial cells. RhoA kinase inhibitor, Y27632, blocks ROCK signaling modulation of the actin cytoskeleton remodeling process[23,24]. Suppressing ROCK signaling with Y27632 inhibitor significantly reduced the size and the occurrence of the intercellular gaps on Au

NPs treated the endothelial cells monolayers (Fig. 2e and Supplementary Fig. 10) and corresponding transwell assays showed significant reduction in the NanoEL levels when the Y27632 inhibitor pre-treated endothelial cells are later exposed to Au NPs (Fig. 2f). Collectively, these observations supported the notion that the rougher Au NPs (R$_3$ groups) increased the activation of VE-cadherin$_{Y658}$ that in turn triggered the internalization and degradation of adherens junction VEC. This consequently depleted the number of VE-cadherin proteins along the adherens junction that are needed to maintain the intact vascular barrier and thereby explain the formation of the gap. Our results highlighted that Au NPs of the higher roughness (R$_3$ to R$_0$) induced the activation of VE-cadherin$_{Y731}$ residue, triggered the dissociation of VE-cadherin and β-catenin, and promoted the actin remodeling process which leads to cell contraction and opening of vascular barrier in a similar mechanism as previously observed[1].

## Nanoparticles increase in vivo tumor vasculature leakiness
As the in vitro monolayer of endothelial cells does not sufficiently represent the complexities of mammalian tumor vasculature, we used live animal intravital imaging to study NanoEL in a tumor microenvironment. We tested three murine cancer models (i.e., slower growing syngeneic 4T1 breast tumor, hypovascular Panc02 pancreatic adenocarcinoma and hyperpermeable CT26 colon adenocarcinoma) which were ectopically implanted into one of the two ear flaps of mice.

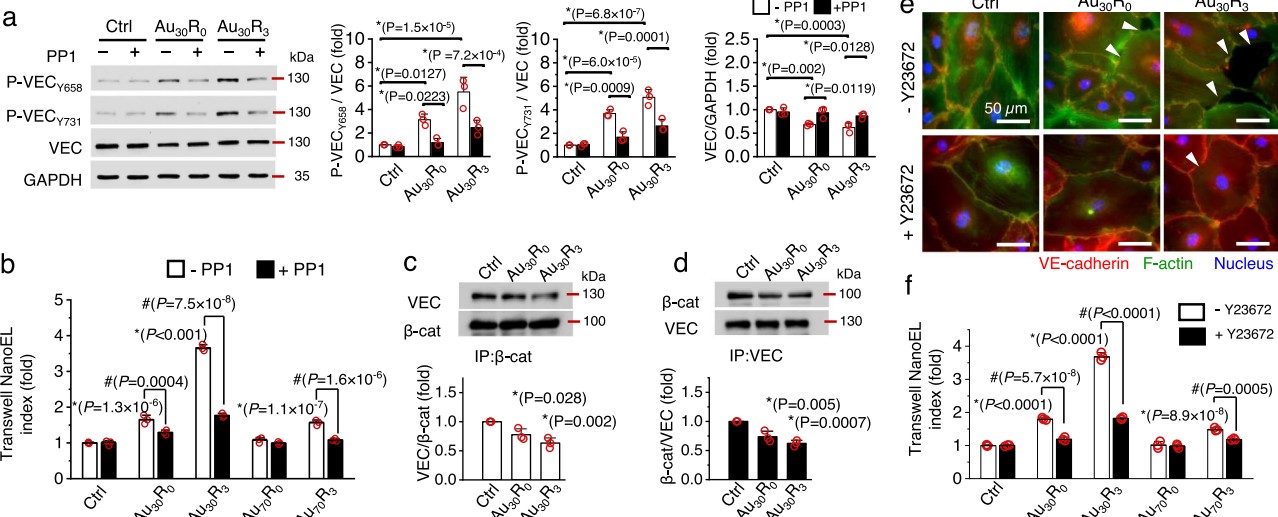

**Fig. 2 | Au NanoEL induction required activation of VE-cadherin signaling and actin remodeling process. a** Immunoblotting and its semi-quantitative analysis show Au$_{30}$ NPs ($1 \times 10^9$ particles/mL, 1 h) induced activation of VE-cadherin (VEC) signaling. Increased phosphorylation of VE-cadherin at tyrosine residues of 658 (P-VEC$_{Y658}$) and 731 (P-VEC$_{Y731}$). Src kinase inhibitor, PP1 (10 μM) effectively repressed the VEC signaling activation. Protein samples of the same experiment were processed in parallel blots. Data are mean ± SD, $n = 3$, One-way ANOVA, Tukey HSD post-hoc test, $P < 0.05$, *significant between compared group. **b** Addition of PP1 reduced the NanoEL index. Data are mean ± SD, $n = 3$, One-way ANOVA, Tukey HSD post-hoc test, $P < 0.05$, *significant between compared groups, #significant against no PP1 treatment. **c** Immunoprecipitation of β-catenin (β-cat) show reduced interaction between VEC and β-cat following Au$_{30}$ NPs

treatment ($1 \times 10^9$ particles/mL, 1 h). **d** The immunoprecipitation of VEC show decreased amount of VEC interacting with its partner, β-cat. **c, d** Data are mean ± SD, $n = 3$, One-way ANOVA, Tukey HSD post-hoc test, $P < 0.05$, *significant against control. **e** Au NPs activated actin remodeling process that led to NanoEL induction. Immunofluorescence images depict reduction of the formed gap on monolayer endothelial cells following ROCK inhibitor, Y27632 treatment (10 μM, 1 h). Scale bar: 50 μm. Nucleus (blue), F-actin (green) and VE-cadherin (red). **f** Y27632 treatment led to reduction in NanoEL. Data are mean ± SD, $n = 3$, One-way ANOVA, Tukey HSD post-hoc test, $P < 0.05$, *significant between compared groups, #significant against no Y-27632 treatment. Source data are provided as a Source Data file.

Au$_{30}$R$_3$ nanoparticles were introduced via tail vein injection into circulation followed by 100 nm green fluorescent DiO-labeled PEG-*b*-PLGA nanoparticles (NPs-DiO) and the latter particles were used as surrogates for any nanoparticles movement within the vasculature of the implanted tumor in the ear flap with live fluorescent high resolution and high-speed intravital imaging (Fig. 3a). Control group showed very little if any entry of NPs-DiO into the interstitial space of the 4T1 ectopic tumor while there was profound leakiness of NPs-DiO into the interstitial space of the tumor in the group pre-treated with Au$_{30}$R$_3$ particles (Fig. 3b, Supplementary Fig. 11, Supplementary Movie 1). The pretreated Au$_{30}$R$_3$ normal vasculature group on the non-tumor bearing ear for the same animal did not show any observable leakiness (Fig. 3c, Supplementary Fig. 11b). Quantification of tumor interstitial efflux of the NPs-DiO both in terms of intensity and interstitial coverage were more pronounced in the Au$_{30}$R$_3$ pretreated group (Fig. 3d, e, Supplementary Movie 2). Consistent with the findings in the 4T1 ectopic tumor model, the images of Panc02 (Supplementary Fig. 12a–c) and CT26 (Supplementary Fig. 13a–c), ectopic tumor vasculatures revealed significant leakiness in the Au$_{30}$R$_3$ pretreated group. This suggests that NanoEL has wide therapeutic effectiveness on various tumor types.

As tumor vasculature varies between locations, we also employed 2 murine orthotopic tumor model, namely 4T1 breast cancer cells (implanted in the mammary pad, Fig. 3f, Supplementary Fig. 14) and Panc02 pancreatic cancer cells (implanted in the pancreas, Supplementary Fig. 15). We found profound leakiness of the tumor vasculatures with Au$_{30}$R$_3$ pretreatment as the case in orthotopic breast (Fig. 3g–i) and orthotopic pancreatic (Supplementary Fig. 15a–c) tumor models. This is consistent with previous observation made with the ectopic models.

Tracking NanoEL occurrence in Au$_{30}$R$_3$ group through the series of time-lapsed images of the 4T1 (Fig. 3j), Panc02 (Supplementary Fig. 12d) and CT26 (Supplementary Fig. 13d) ectopic tumor

vasculatures as well as the 4T1 (Fig. 3k) and Panc02 (Supplementary Fig. 15d) orthotopic tumor vasculatures, we observed an accumulation of the NPs-DiO particles which appeared to be a slight aneurysm-like phase of the tumor vessels followed by a burst of leaked NPs-DiO over an order of minutes (Fig. 3j, k, Supplementary Fig. 12d, Supplementary Fig. 13d, Supplementary Fig. 15d). In line with our observation, the pioneering modelling work by Netti et al. demonstrated the relief of tumor interstitial pressure could result in increased drug delivery to the tumor[25]. Furthermore, Matsumoto et al., gave visual evidence of the extravasation of the particles that followed the convection forces manifesting as vascular blood pressure bursts[26]. Thus, there is a possibility that both vascular bursts and NanoEL effect work in unison to increase the leakiness phenotype where the latter is of a controllable nature with nanotechnology while the former is the inherent pulsatile pressure of our cardiovascular system.

One of the fundamental questions that would need to be addressed in order to realize the therapeutic application of NanoEL is whether the induction is a permanent or temporal phenomenon. Utilizing IVM analysis on ectopic 4T1 breast cancer model over the course of 10 hours we observed that the NanoEL induction was transient in nature (Supplementary Fig. 16). Leakiness, evidenced by the presence of DiO-NPs tracers in the tumor interstitial space, was noted as early as 70 min post tracer injection and persisted at least for another 440 min (~ more than 7 h) before dissipating back to the basal level 540 min post tracer injection. We previously demonstrated in in vitro endothelial monolayer model that TiO$_2$ NPs were able to induce similar leakiness for at least 6 h[2]. In contrast, thrombin, a well-known inducer of vascular permeability is only effective opens up vasculature for 2.5 h[27]. This showcases the therapeutic window which could utilize to facilitate nanomedicine delivery.

Next, we attempted to treat primary tumor cancer by increasing therapeutic access to the tumor via first using NanoEL nanoparticles.

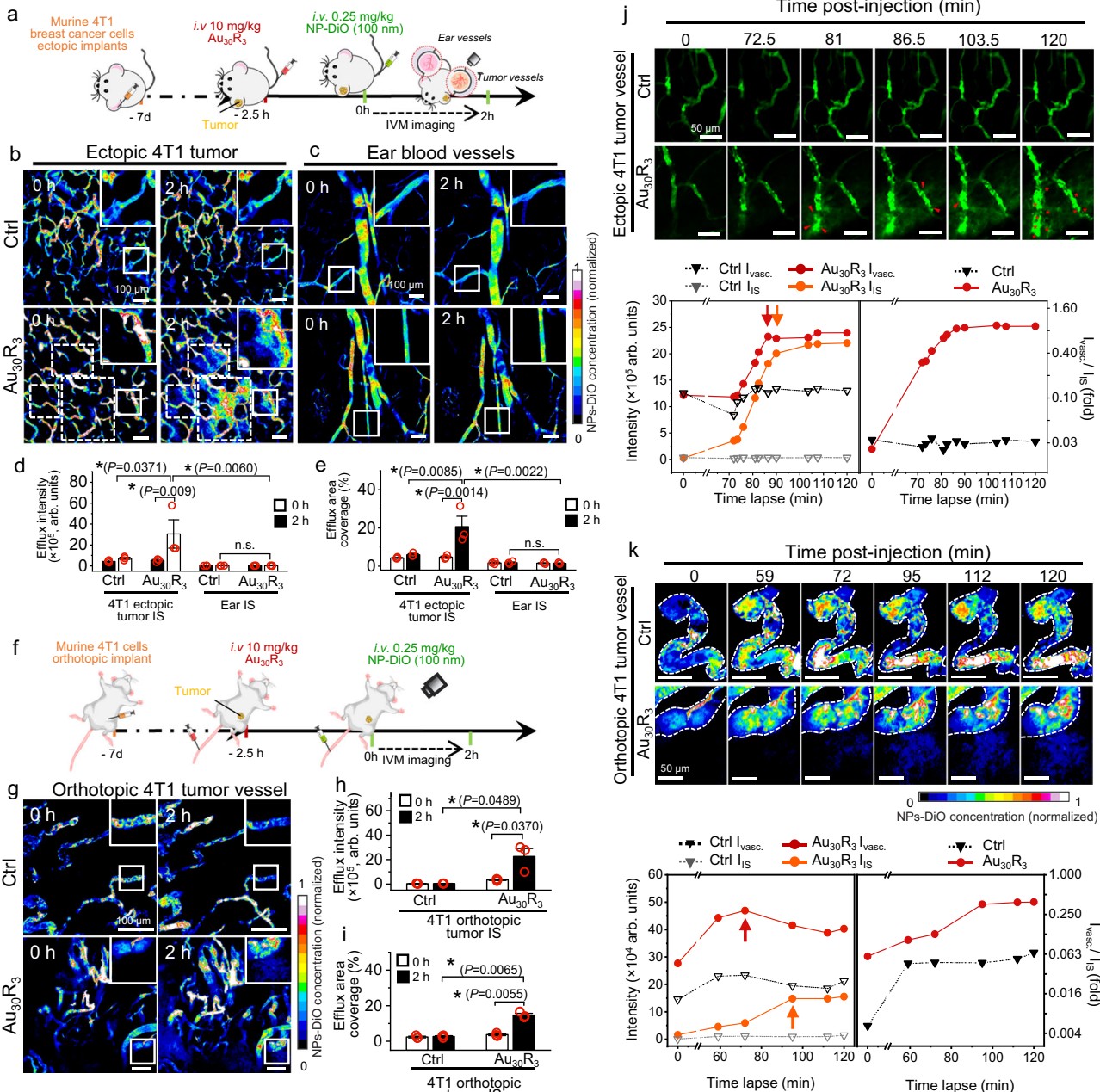

**Fig. 3 | NanoEL increased leakage of blood borne entities into the interstitial space of the tumor. a** Schematic of the animal intravital imaging (IVM) of the ear flap vasculature ectopically implanted with 4T1 breast tumor cells ($n = 3$ mice/group). Multicolor images show increased leakiness into the interstitial space in the (**b**) 4T1 tumor model in the $Au_{30}R_3$ nanoparticles group. **c** No observable leakiness occurred at the other ear blood vessels of the same animal despite subjected to the same tail vein intravenuous introduction of $Au_{30}R_3$. Scale bar: 100 μm. **d** Quantification of the efflux intensity and **e** area of the interstitial area showed that $Au_{30}R_3$ has caused profound leakiness. Data are mean ± SEM, $n = 3$ mice/group. Two-way repeated measures ANOVA Tukey post-hoc test, $P < 0.05$, *significant between compared groups. **f** Schematic of the IVM of the orthotopically implanted murine breast tumor cells 4T1. $n = 3$ mice/group. **g** Multicolor images show increased leakiness into the

interstitial space in 4T1 orthotopic tumor vessels in the $Au_{30}R_3$ nanoparticles group. Scale bar: 100 μm. Quantification of the **h** efflux intensity and **i** area of the interstitial area showed that $Au_{30}R_3$ has caused profound leakiness. Data are mean ± SEM, $n = 3$ mice/group, Two-way repeated measures ANOVA Tukey post-hoc test, $P < 0.05$, *significant between compared groups. **j, k** Time lapse evolution of leakiness of a focused area in (**j**) 4T1 ectopic tumor vasculatures, and (**k**) 4T1 orthotopic tumor vasculatures. Initially, due to a weakened vessel wall arising from NanoEL, there appeared to be aneurysm events that led to further leakage. Profiling of NPs-DiO particles load in the vasculature versus interstitial space. The load peaks earlier in the vasculature (red arrow) than in the interstitial space (orange arrow). Scale bar: 50 μm. **j, k** The images shown are representative of three independent experiments. Source data are provided as a Source Data file.

---

Subcutaneously implanted 4T1 tumors on the flanks (ectopic) or into the mammary fat pad (orthotopic) of the mice (Fig. 4a). The mice were pre-treated with $Au_{30}R_3$ nanoparticles or $Au_{70}R_0$ nanoparticles or vehicle control with a second injection of a larger 100 nm red fluorescent NPs-DiD marker to visualize the leakiness (Fig. 4a). If there is induced leakage, one would expect sustained accumulation of the

leaked NPs-DiD from the blood vessel into the interstitial space and remained there. $Au_{30}R_3$ pretreatment resulted in the fastest rate of accumulation and remained the highest level amongst the $Au_{70}R_0$ and control groups (Fig. 4b, c, Supplementary Fig. 17). After 24 h, the animals were sacrificed, and the fluorescence of the various organs were measured. As expected, even after 24 hours, the leaked nanoparticle

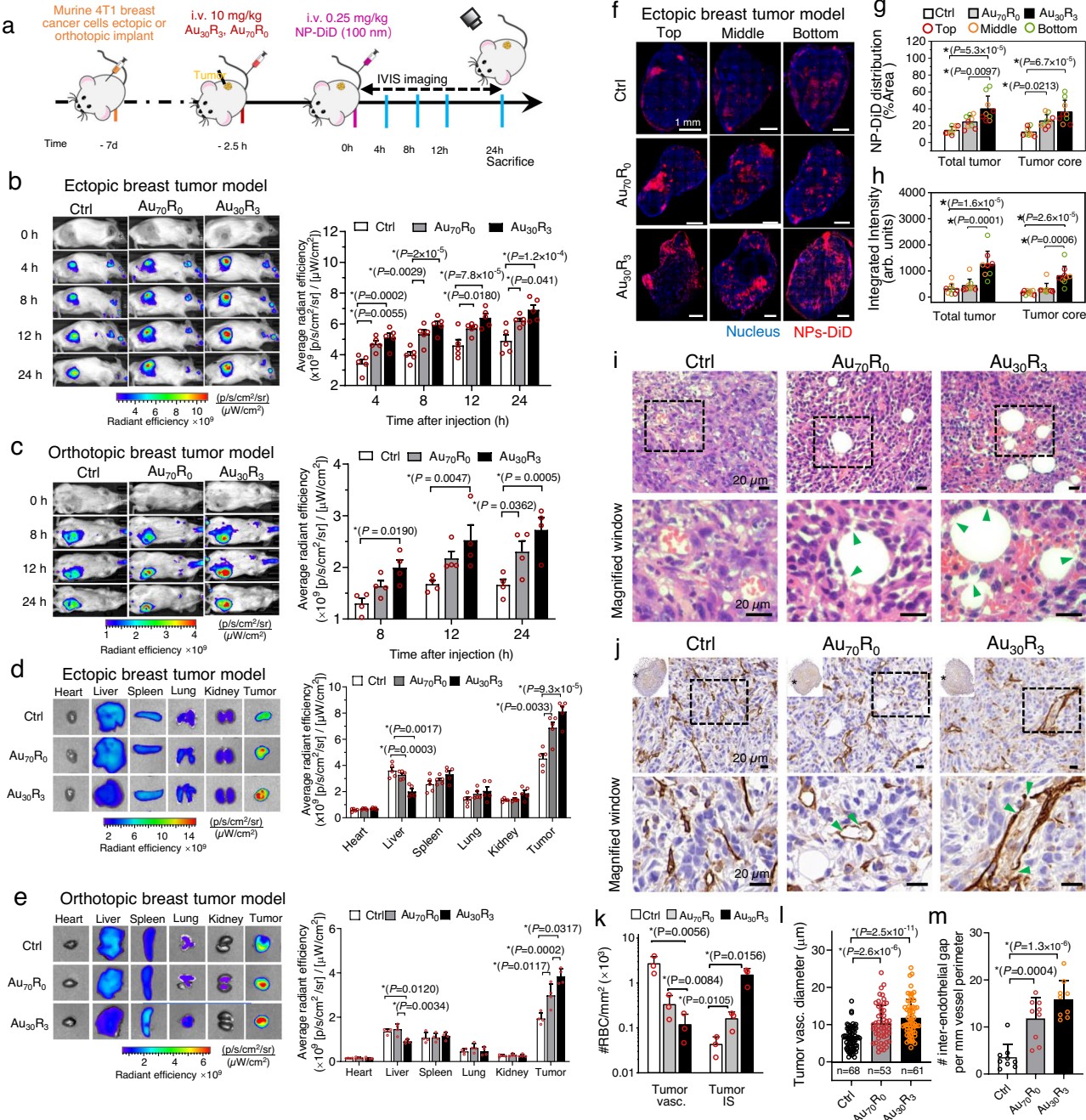

**Fig. 4 | NanoEL particles increase vascular leakiness resulting in deeper tumoral penetration. a** Mouse treatment scheme. Live animal imaging on **b** ectopic ($n = 5$ mice/group) and **c** orthotopic breast 4T1 ($n = 4$ mice/group) tumors up to 24 h. Faster and higher accumulation of fluorescent NPs-DiD at the tumor in the NanoEL particles groups ($Au_{30}R_3$ and $Au_{70}R_0$) compared to the vehicle-control. Saturation is quickly reached and persisted in $Au_{30}R_3$ group. Two-way repeated measures ANOVA Tukey HSD post-hoc test. After 24 h, animals bearing **d** ectopic ($n = 5$ mice/group) and **e** orthotopic ($n = 4$ mice/group) 4T1 tumors were sacrificed; organs were scanned for fluorescence. $Au_{30}R_3$ group - highest tumoral accumulation of fluorescent particles with the lowest hepatic accumulation. One-way ANOVA Tukey HSD post-hoc test. **f** Ectopic tumor sections shows little NPs-DiD infiltration in the control group and significantly higher tumoral penetration of NPs-DiD in $Au_{70}R_0$ and $Au_{30}R_3$ groups. Scale bar: 1 mm. Quantification of the NPs-DiD (**g**) coverage area and (**h**) intensity in the representative images of ectopic 4T1 tumor sections ($n = 9$ sections, $N = 3$ mice/group). One-way ANOVA Tukey HSD post-hoc

test. **i** H&E stained 4T1 ectopic tumor control showed RBCs were confined within tumor vasculature, while in Au NPs treated group, the RBCs were found in the interstitial spaces. Scale bar: 20 μm. **j** CD31 immunohistochemistry (IHC) staining of the cross-section of the orthotopic 4T1 tumor show leakiness in tumor vasculature in Au NPs treated groups. Tumor vasculature in Au NPs treated groups was distended with occurrences of gaps (green arrows) around the circumferential regions. Scale bar: 20 μm. **k** RBC counts in the blood vessels and the interstitial spaces. $n = 3$ mice/group, One-way ANOVA Tukey HSD post-hoc test. **l, m** Quantification of the tumor vasculature cross-sectional diameter of the Au NPs treated groups showed (**l**) a higher mean diameter over that in the control tumor vasculature group ($n = 3$ mice/group) and (**m**) increased number of inter-endothelial gaps ($n = 9$ IHC staining, $N = 3$ mice/group). One-way ANOVA Tukey HSD post-hoc test. **b–e** Data are mean ± SEM. **g–h, k–m** Data are mean ± SD. **b–e, g, h, k–m** *significance against control $P < 0.05$. Source data are provided as a Source Data file.

tracers remained within the tumor that were pretreated with $Au_{30}R_3$ nanoparticles which further emphasized that it is accumulating through leakage and not merely through persistent residence time in the blood vessels (Fig. 4d, e, Supplementary Fig. 18). To further validate this, we employed orthotopic pancreatic tumor model and utilized both IVIS imaging (Supplementary Fig. 19a) and fluorescence reflectance imaging (Supplementary Fig. 19b) to assess the accumulation of tracer particles NPs-DiD. Consistent with the earlier described observations, Au NPs pretreatment resulted in the highest accumulation in the tumor (Supplementary Fig. 19).

Comparing the tracer accumulation levels of the various organs and tumors of the same animal, when pretreated with NanoEL $Au_{30}R_3$ particles group, the tumor showed the highest leakage tracer accumulation. It is surprising as the liver, spleen, and kidney are the usually reported sites of nanoparticles accumulation (Fig. 4d, e, Supplementary Figs. 18, and 19). Validation with ICP MS supported the increase of Au NPs in tumor tissues as compared to other organs (Supplementary Fig. 20). Considering that we have used large tracer particles which will not be quickly cleared by the liver and kidney suggest that the accumulation in the tumor is indeed very high. So, if one were to replace these tracer particles with therapeutics, a similar degree of tumor accumulation and depressed non-tumor organs accumulation would be expected. This suggests that if robust leakiness of the tumor vasculature can be achieved, the tumor might become the therapeutics sink and in essence reduced the anti-cancer therapeutic disposition to the rest of the healthy organs and reduced the off-target toxicity and side effects. Consistent with this observation, representative entire tumor cross-sections of 4T1 ectopic tumor (Fig. 4f, Supplementary Fig. 21), 4T1 orthotopic tumor (Supplementary Fig. 22a), and the pancreatic orthotopic tumor (Supplementary Fig. 23a) at the three section planes revealed that control group showed little penetration of the NPs-DiD into the interior sections of the tumor. In contrast, $Au_{70}R_0$ and $Au_{30}R_3$ treated groups showed significantly higher and deeper penetration of NPs-DiD particles to the core of the tumors (Fig. 4g, h, Supplementary Fig. 22b and Supplementary Fig. 23b).

Hematoxylin and eosin (H&E) histology (Fig. 4i, Supplementary Fig. 24) and CD31 immunohistochemistry staining (Fig. 4j, Supplementary Fig. 25) of the tumor vasculatures reveal that without Au NPs pre-treatment, there was no discernible leakiness in the tumoral vasculature with red blood cells still present in the blood vessels. In contrast, in the Au NPs pre-treated group, most of the red blood cells were found the interstitial regions and less in the vasculature (Fig. 4i, k). The in vivo NanoEL gaps are estimated to be even as wide as 10 microns in size as red blood cells could leak from the vasculature. Au NPs treatment appeared to result in distended vasculatures with a significant increase in the mean diameter (Fig. 4l) of the vasculatures in addition to the increase in the average of the measured cross-sectional area of the tumor vasculatures which get enlarged by 27% and 45% following $Au_{70}R_0$ and $Au_{30}R_3$ pretreatment, respectively (Supplementary Fig. 25). Assuming that the circumferential plane tumoral vasculature across all the groups before treatment is formed by the same number of endothelial cells, the enlarged vasculature in the Au NPs treated group would have to mean that the endothelial cells were stretched or as the result of gaps formation (green arrows Fig. 4i, j). Additionally, we noted a significant increase in the number of visible gaps between the endothelial cells in these distended blood vessels (Fig. 4m, Supplementary Fig. 25).

We noted no cytotoxic effect in the mice receiving the Au NPs treatment (Supplementary Fig. 26). Hematoxylin and eosin (H&E) histology (Supplementary Fig. 26a) of the normal organs (i.e., heart, liver, spleen, kidney, and lung) showed no perceivable morphological difference after $Au_{70}R_0$ and $Au_{30}R_3$ treatment. Considering that these normal organs receive almost the entire volume of blood that carries the Au nanoparticles, if there was any robust leakiness, it should result in a discernible degree of leakage of NP-DiD from the vasculature

and accumulation in some or all of the organs. Additionally, no abnormality could be associated with NPs treatments with regards to the white blood cell, red blood cell counts, blood biochemical analysis (Supplementary Fig. 26b–e), as well as liver and kidney function (Supplementary Fig. 26f–h). CD31 IHC staining of the normal organs (i.e., liver, spleen, and kidney) vasculatures (Supplementary Fig. 27) showed no detectable Au NPs induced leakiness at these organs over non-treated controls. An additional control experiment (Supplementary Fig. 28a) employing no tumor-bearing mice shows that Au NPs did not result in discernible differences in the DiDs accumulation in the normal organs (Supplementary Fig. 28b, c).

As NanoEL in the in vitro setting was found to be initiated through the VE-cadherin signaling (Fig. 2), we, therefore, examined whether the same signaling plays a pivotal role in the induction of NanoEL in vivo. Consistent with the previous observation made through CD31 IHC staining (Fig. 4j), we noted increase in the intercellular gaps (green arrows) around the circumferential region of the tumor vasculature in the Au NPs (Supplementary Fig. 5a and Supplementary Fig. 29). VE-cadherin (VEC) IHC staining on the cross sections of 4T1 orthotopic tumor vasculatures (Fig. 5a and Supplementary Fig. 29) indicated significant reduction in the VEC expression around the circumferential region of the tumor vasculature in the Au NPs treated groups, with the largest reduction (~close to 20% reduction) was noted in the $Au_{30}R_3$ treated group (Fig. 5b). In concomitant, significant increase phosphorylation of VE-cadherin at Y658 residue ($P\text{-VEC}_{Y658}$) around the circumferential region of the tumor vasculature was noted in the Au NPs treated groups (Fig. 5a, b). The results are in line with our in vitro observation (Fig. 2) and provide an early indication that VE-cadherin activation is the major driver of in vivo NanoEL.

To further validate the involvement of VE-cadherin signaling in vivo, we introduced PP1 (1.5 mg/kg) treatment 1 hour prior to introducing $Au_{30}R_3$ particles. PP1 treatment has been demonstrated to significantly reduce the level of phosphorylated VE-cadherin in the in vitro study (Fig. 2a and Supplementary Fig. 9a), resulting in a significant reduction of NanoEL levels (Fig. 2B). Therefore, blocking VE-cadherin signaling with Src inhibitor PP1 is expected to result in the normalization of endothelial integrity in the in vivo setting. We utilized mice implanted with 4T1 ectopic tumor on their ear flaps. Pre-treating the mice with PP1 alone did not change the leakage observed in the 4T1 ectopic tumor (Fig. 5c, d, Supplementary Fig. 30). Consistent with the previous intravital imaging data (Fig. 3b), we noted significant leakage of tracer particle NPs-DiO into the interstitial space in $Au_{30}R_3$ group. This leakiness was significantly attenuated with pretreatment PP1 (Fig. 5c, d, Supplementary Fig. 30). There was more accumulation of tracer NPs-DiD in the 4T1 orthotopic tumor in the group with $Au_{30}R_3$ pretreatment compared to the control treatment. This increased accumulation was abrogated with the PP1 treatment of the mice (Fig. 5e, f, and Supplementary Fig. 31a). IVIS imaging and quantification of the signal on the harvested organs also supported the role of VE-cadherin axis in vivo NanoEL where PP1 again abrogated the otherwise increased $Au_{30}R_3$ pretreatment induced accumulation of the tracer in the tumor (Fig. 5g, h, and Supplementary Fig. 31b).

## NanoEL improves therapeutic outcomes in primary tumors and metastasized cancer

Next, we hypothesized that if we were able to increase in vivo vascular leakiness, small molecule drugs, nano therapeutic carriers, vaccine carriers will be able to exploit NanoEL increased access to better penetrate the tumor and exert their therapeutic effects. We investigated two important tumor scenarios. The first is treatment of the primary tumor; the second, addressing micrometastases treatment.

In the primary tumor treatment study, doxorubicin (Dox), which on its own has limited therapeutic efficacy against early-stage 4T1 tumors at the earlier time points[28], was used as a control in addition to its more efficacious liposomal Dox (Lipo-Dox) to investigate the effect

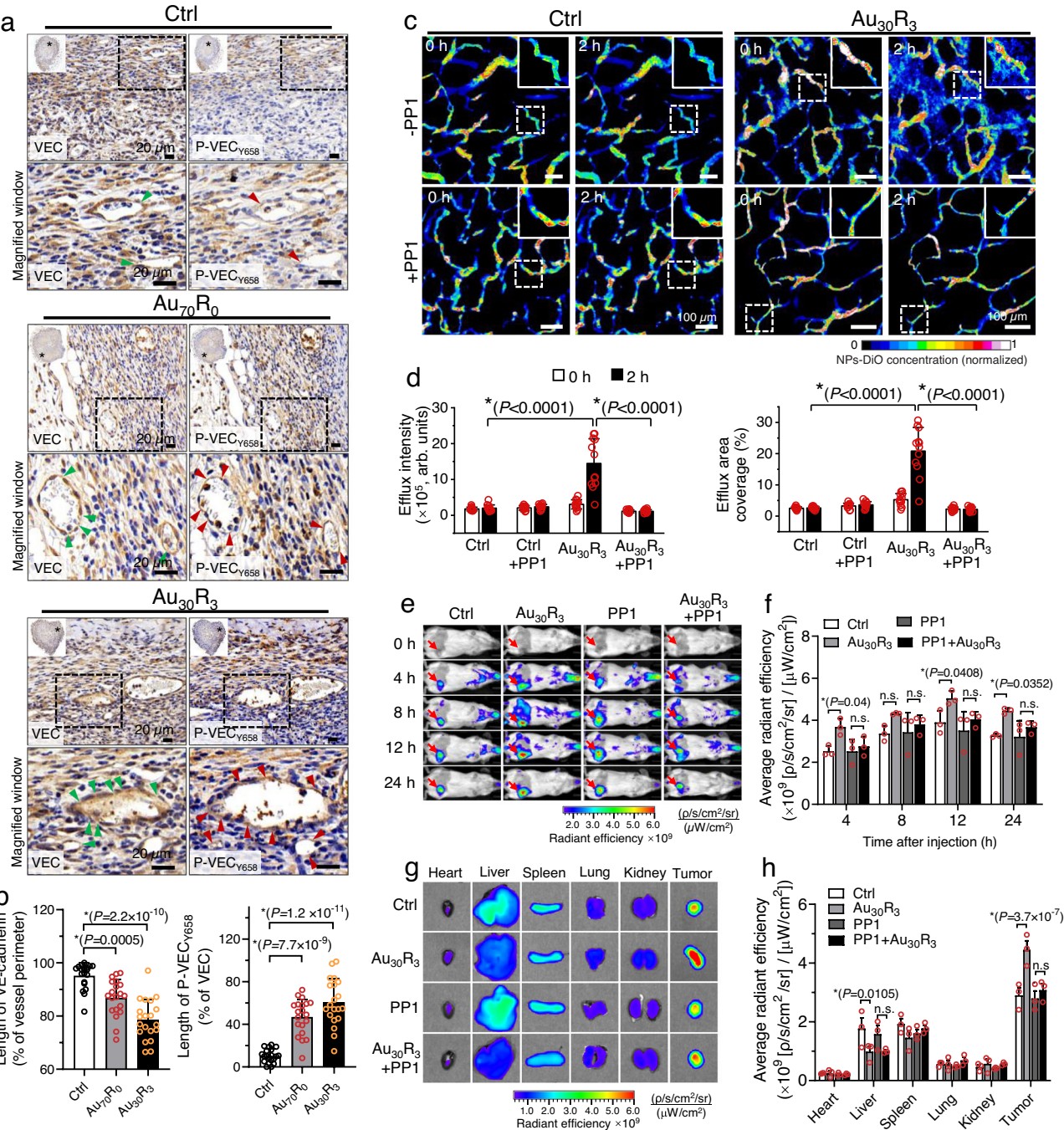

**Fig. 5 | VE-cadherin signaling is required in NanoEL induction in vivo. a, b** Au NPs treatment activates the VE-cadherin signaling in the in vivo setting. **a** Bright field images and **b** and quantification of IHC of VE-cadherin (VEC) and phosphorylated VE-cadherin (P-VEC$_{Y658}$) on the 4T1 orthotopic tumor vasculature cross sections. IHC staining shows reduced VEC expression of tumor vasculature of Au NPs treated groups with prominent gaps (green arrows) observed at the circumferential region of tumor vasculature. Concomitantly, IHC staining shows increased cell expression of P-VEC$_{Y658}$ (red arrows). Scale bar: 20 μm. $n = 9$ independent IHC staining, $N = 3$ mice/group. Data are mean ± SD, One-way ANOVA Tukey HSD post-hoc test. **c, d** Inhibiting VE-cadherin signaling through PP1 (Src kinase inhibitor) pre-treatment reduced NanoEL observed in 4T1 ectopic tumor vasculature. Ectopic murine 4T1 breast tumor cells implanted in mouse ear flap ($n = 3$ mice/group). Mice received PP1 pretreatment (1.5 mg/kg, 1 h) prior being treated either with vehicle control or Au$_{30}$R$_3$ (10 mg/kg). **c** IVM images of ear flaps vasculature with ectopic 4T1 tumors ($n = 3$ mice/group). Scale bar: 100 μm.

**d** Fluorescent quantification of efflux intensity and interstitial area show PP1 pre-treatment significantly attenuate Au$_{30}$R$_3$ NPs induced leakiness. Data are mean ± SD, $n = 12$ IVM images/group, $N = 3$ mice/group. Two-way repeated measures ANOVA Tukey post-hoc test. **e**–**h** Inhibiting VE-cadherin signaling through PP1 treatment reduced NanoEL observed in 4T1 orthotopic tumor vasculature. Ortho-topically 4T1 tumor cells implanted on the mouse ($n = 3$ mice/group). Mice received PP1 pretreatment (1.5 mg/kg, 1 h) prior being treated either with vehicle control or Au$_{30}$R$_3$ (10 mg/kg). **e** IVIS imaging of live animal implanted with orthotopic 4T1 tumors ($n = 3$ mice/group). **f** PP1 pre-treatment reduced the accumulation of NPs-DiD in the tumor over the course of 24 h. Data are mean ± SEM, Two-way repeated measures ANOVA Tukey HSD post-hoc test. **g** IVIS imaging of various organs ($n = 3$ mice/group) harvested from the mice. **h** Fluorescent quantification showed reduced leaked tracer following PP1 treatment. Data are mean ± SEM, One-way ANOVA Tukey HSD post-hoc test. **b, d, f, h** *significance against control $P < 0.05$. Source data are provided as a Source Data file.

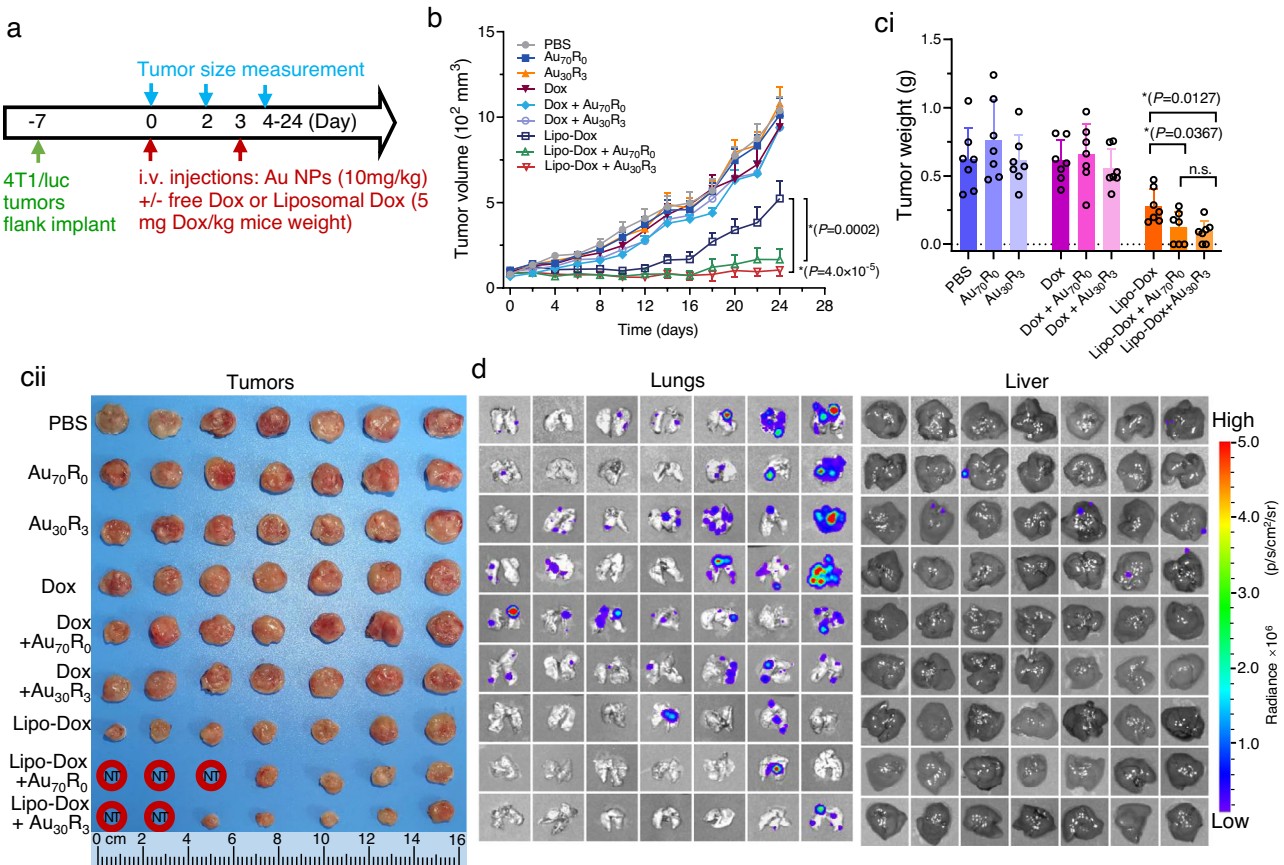

**Fig. 6 | NanoEL particles improves nano drug carrier access and therapeutic outcomes in early-stage tumor. a** Mice treatment scheme. NanoEL particles: $Au_{30}R_3$ and $Au_{70}R_0$. $n = 7$ mice/group. **b** Tumor size profile over 24 days of treatment. The largest tumor regression and lasting effect was seen in the case of liposomal Dox delivery groups with almost complete regression of the tumors when there was treatment with NanoEL Au NPs. Data are mean ± SEM, $n = 7$ mice/group, Two-way repeated measures ANOVA, Tukey HSD post-hoc test, *significant against Lipo-Dox group, $P < 0.05$. **c** After 24 days, the animals were injected with luciferin and sacrificed. The whole tumors were excised and visually size-sorted. (ci) Quantification data and (cii) tumor images show the Lipo-Dox alone treatment showed significant regression of the tumors but the most significant regression is found

those with NanoEL Au NPs co-treatment. Some of the mice have complete regression of their tumors (NT – no tumor found). Data are mean ± SD, $n = 7$ mice/group, One-way ANOVA, Tukey HSD post-hoc test, *significant against Lipo-Dox group, $P < 0.05$. **d** IVIS scanning of lungs and livers from the same mice shows the presence of metastatic 4T1/luc. NanoEL Au NPs treatment without Dox or Lipo-Dox increases lung and liver metastasis. Au NPs co-treatment with Dox or Lipo-Dox drastically reduced the metastasis to lung and liver metastasis. For the five tumors that were completely regressed, there was correspondingly no signs of metastasis. Other excised organs did not show observable metastasis (Supplementary Fig. 33). Source data are provided as a Source Data file.

of adjusting the in vivo vascular leakiness. We treated implanted 4T1/luc tumors with $Au_{30}R_3$ or $Au_{70}R_0$ with either small molecule drug Dox, or the larger ~75 nm Lipo-Dox for only two-time points (Day 0 and Day 3 of the treatment regime) (Fig. 6a). Subsequently, tumor size measurements (Fig. 6b) and mice weight (Supplementary Fig. 32a) were recorded until Day 24 when the animals were sacrificed. The tumors of most groups including free Dox, with or without Au NPs did not show final regression of the tumor compared to no-treatment control (Fig. 6B). At the earlier time points right after treatment with even the free Dox in the Au NPs groups versus free Dox alone, but without sustained and repeated Dox and Au NPs treatment, surviving cancer cells bounced back and continued their upward growth trajectory (Fig. 6b). The more potent Lipo-Dox at the same Dox concentration significantly slowed the tumor growth down (Fig. 6b). However, if we increase the vascular escape with NanoEL Au NPs and with better penetration into the tumor (Figs. 3 and 4) at the beginning, Lipo-Dox now will be better able to exert its therapeutic effect, tumor sizes diminished (Fig. 6b). That initial and only burst of increased NanoEL effect and therapeutic effects was sufficient in reducing tumor sizes and in some mice, were completely cured of the tumors (Fig. 6c). It is worth noting that even though the treatment were only carried out in

the earlier time points of Day 0 and Day 3 with therapeutics that is not the most potent in its class, it could achieved complete tumor regression. This further accentuates the improvements that earlier increased vascular leakiness with our nanotechnology can greatly potentiate the cancer therapeutics' effect.

In clinical settings, late stage tumors are exceedingly difficult to treat and typically results in the poor prognosis[29,30]. We modified the ectopic breast tumor model in which the primary tumor was left to grow to create inoperable tumor model (of ~500 mm³ tumor volume)[30] and once again investigated the effect of adjusting the in vivo vascular leakiness on the efficacy of Lipo-Dox therapy. The 4T1/luc large tumors were treated with $Au_{30}R_3$ or $Au_{70}R_0$ and Lipo-Dox formulation for only two-time points (Day 0 and Day 6 of the treatment regime) (Fig. 7a), and the tumor size measurements (Fig. 7b) and mice weight (Supplementary Fig. 32b) were recorded until Day 12 when the animals were sacrificed. Similar to the observation on the early-stage tumor model (Fig. 6), in the absence of Lipo-Dox formulation, Au NPs groups did not show final regression of the tumor compared to no-treatment control (Fig. 7b–d). Lipo-Dox only formulation was demonstrated to once again significantly slowed down the growth of the late-stage tumor (Fig. 7b); whereas its co-treatment with NanoEL $Au_{70}R_0$ NPs was noted

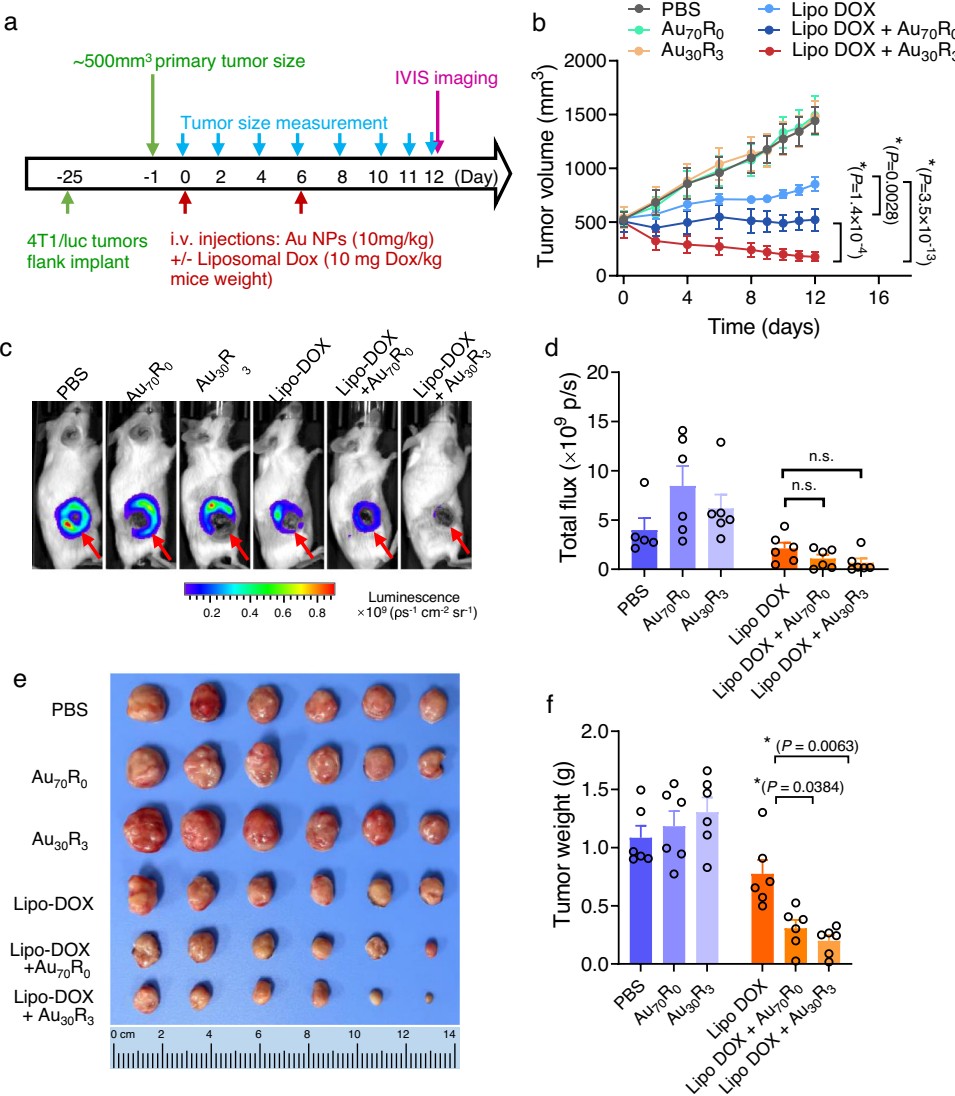

**Fig. 7 | NanoEL particles improves nano drug carrier access and therapeutic outcomes in late-stage tumor. a** Mice treatment scheme. NanoEL particles: $Au_{30}R_3$ and $Au_{70}R_0$. $n = 6$ mice/group. Treatment was initiated when the tumor reached the volume of ~500 mm³, and only carried out on Days 0 and 6. **b** Tumor size profile throughout the entire 12 days of treatment. Au NPs treatment appeared to improve liposomal Dox delivery groups. Data are mean ± SEM, $n = 6$ mice/group, Two-way repeated measures ANOVA, Tukey HSD post-hoc test, *significant against Lipo-Dox group, $P < 0.05$. After 12 days, the animals were injected with luciferin and the 4T1/luc tumor was visualized using the IVIS system before being sacrificed, and the whole tumors were harvested and size-sorted. **c** Representative IVIS images and **d** Quantification of total flux show reduce tumor size in the animal receiving liposomal Dox and Au NPs group. Data are mean ± SEM, $n = 5$ mice for PBS control group, $n = 6$ mice/group for the other treatment groups. One-way ANOVA, Tukey HSD post-hoc test. **e** Tumor images and **f** quantification data show the Lipo-Dox alone treatment showed significant inhibition of the late-stage tumor growth, but most significant regression is found in those with NanoEL Au NPs co-treatment. Data are mean ± SEM, $n = 6$ mice/group. One-way ANOVA, Tukey HSD post-hoc test, $P < 0.05$, *significant against Lipo-Dox group. Source data are provided as a Source Data file.

to stay the growth of this late-stage tumor. More interestingly, increasing the vascular escape with NanoEL $Au_{30}R_3$ NPs was demonstrated to effectively lead to late tumor regression (Fig. 7b). This further supports that the combined effects of increased NanoEL and Lipo-Dox formulation were effective in reducing tumor sizes (Fig. 7e, f), highlighting that increased vascular leakiness with our nanotechnology can greatly potentiate the cancer therapeutics' effect even introduced at later stage of the tumor progression.

The classical alternative view of increased tumoral vascular leakiness can lead to increased metastasis and increased nutrient supply to the tumor and thus opposes the clinical outcome that the cancer therapy is trying to achieve[2]. Our observations supported this view where increased metastasis was observed in the $Au_{30}R_3$ and $Au_{70}R_0$ groups without the cancer therapeutics treatment compared to the

control groups (Fig. 6d, Supplementary Fig. 33). Conversely, when there is efficacious access of anti-cancer therapeutics to the tumor through NanoEL significantly reduced this metastatic phase and in some cases, absent (Fig. 6d, Supplementary Fig. 33). Thus, NanoEL provided a short window of therapeutic opportunity; temporally long enough for the much faster and leakage of therapeutics to effectively access the tumor but not temporally long enough for cancer cells to become metastatic and migratory to exploit this route of escape.

## NanoEL improves therapeutic outcomes in metastasized secondary tumors

Tumor metastasis accounts for the majority of cancer deaths. Most clinical cancer practices would surgically remove the primary tumor

but unfortunately could not repeat the same procedure with the secondary micrometastases for late-stage cancers. As these micrometastases have not reached the critical size to induce the EPR effect and despite being far fewer in cell numbers compared to secondary tumors, the micrometastases remained hard to kill without an infiltrating and leaky vasculature for nanomedicine access. This inability to access and eradicate the secondary metastasized tumor may have encouraged therapy to be relegated to a stage where the secondary tumor is large enough to have a leaky vasculature of which it would be too late. Thus, we questioned whether NanoEL can also provide increased therapeutic access to micrometastases colonies.

We modified the orthotopic breast tumor model in which the primary tumor was left to grow without any therapeutic intervention. Based on the earlier mouse studies, we could work backwards to estimate the approximate time of metastasis occurrence and in this mouse experiment, we first excised the primary tumors out to align the study to the clinical practice and to decouple the already metastasized colonies from potentially new metastases. We utilized 4T1-EGFP tumor cells which have been known to metastasize to the lung, we specifically zoom into the lung to look for these metastasis colonies. We search intravitally the mice lungs to find the 4T1-EGFP micrometastases (Supplementary Fig. 34a). Utilizing CD31 counterstaining to visualize the vasculature, it could be observed there was little vasculature infiltrating within the micrometastases colony. Despite the absence of neovascularization, we noted that the $Au_{30}R_3$ NPs treatment could induce accumulation of NPs-DiD in the micrometastasis site (Supplementary Fig. 34b, c). This suggests that NanoEL indeed could provide increased therapeutic access to micrometastases colonies.

To further validate the data, we repeated the experiment by employing the late-stage tumor mode, excising the primary tumor, and treating the mice with $Au_{30}R_3$ or $Au_{70}R_0$ with either small molecule drug Dox, or the larger ~75 nm Lipo-Dox for four time points (Day 0, 4, 8 and 12) of treatment regime (Fig. 8a). With IVIS live animal imaging, we tracked the locations of any significant 4T1/luc breast cancer micrometastases. Both Dox and Lipo-Dox with Au NPs cotreatment managed to kill off the micrometastases with the latter combination showing the mice to be completely micrometastases-free (Fig. 8b). Analysis of the visible lung metastatic foci at the end of the treatment and observation regime (Day 15) showed cotreating the mice with Lipo-Dox and Au NPs could effectively reduce the number of metastatic foci on the lungs with $Au_{30}R_3$ + Lipo-Dox treatment group gave the most significant reduction in the micrometastasis (Fig. 8c). This was also supported by the lung histology sections which is a common metastasis site for breast cancer. Mice receiving cotreatment of Lipo-Dox and Au NPs were noted to have significantly reduced degree of metastatic cancer colony infiltrating their lungs and in specific lungs, was metastases-free (Fig. 8c, d, Supplementary Fig. 35) as in the $Au_{30}R_3$+Lipo-Dox case.

The issue of inducing leakiness with nanoparticles in a tumor vasculature context is a double-edged sword. In the therapeutic study (Figs. 6–8), the NanoEL step and cancer therapeutics step can be experimentally administered where the tumor is in the earlier stages of the tumor progression and therefore, we can temporally segregate the NanoEL from the metastatic step to achieve an Goldilocks window of opportunity for this NanoEL driven therapy. Timing of the NanoEL effect can be tricky in the real clinical step. But we reasoned that since, in vitro endothelial cells studies, the NanoEL window is in the vicinity of hours[2], while the arousal of pro-migratory metastatic cancer cells from amongst the primary tumor cells population are in the realm of weeks and months, we can then coordinate the therapeutic intervention with the NanoEL induction to predictably engineer gaps that are fast but transient such that therapeutics can escape from the deeply penetrating tumor vasculature, accumulate in the interstitial space for longer periods and exert their therapeutic killing of the tumor cells before they turn metastatic. Our strategy of keeping the NanoEL

frequency as low as possible created that Goldilocks situation for maximally effective therapeutic strikes at the tumor.

In summary, we have shown that in a leakiness-deficient tumor vasculature, NanoEL Au NPs could increase their leakiness of nanoparticles. Synergizing NanoEL with the therapeutic efficacy of liposomal Dox led to complete regression of the tumors despite the minimal dosing frequency. In vivo NanoEL focusses on already weakened vessel walls in the tumor vasculature while sparing healthy vasculature. NanoEL gaps help cancer therapeutics better reap the anti-cancer therapeutic benefits. Nanotechnology through the NanoEL effect had provided the independent much needed endothelial escape without depending on the EPR effect originating from the tumor.

## Methods

### Animal care
Female BALB/c mice (6–8 weeks) and male C57BL/6 mice (6–8 weeks) were purchased from Beijing Vital River Laboratory Animal Technology (China). All mice used in this study were housed in a specific pathogen-free facility and received care in compliance with the guidelines outlined in the Guide for the Care and Use of Laboratory Animals. Animals were maintained at $21 \pm 1$ °C, in 40% to 70% humidity, and with a 12 h light/dark cycle (from 8 a.m. to 8 p.m.). All mouse experiment protocols (USTCACUC1801006) were approved by the University of Science and Technology of China Animal Care and Use Committee. The maximum tumor burden permitted is 2000 mm³, and the maximal tumor size/burden was not exceeded in the experiments. Experimental group sizes were approved by the regulatory authorities for animal welfare after being defined to balance statistical power, feasibility, and ethical aspects. The sex of the animal was not specifically considered in this study. Mice of the appropriate sex were employed according to the requirements of the tumor model. A female mouse model was employed for 4T1 breast cancer and CT26 colon cancer, while a male mouse model was used to establish Panc02 pancreatic cancer model.

### Cell culture
Primary human microvascular endothelial cells (HMVECs) were obtained from ThermoScientific (USA; Cat# C-010-5C) and were cultured in growth medium EndoGRO-MV-VEGF (Merck Millipore, USA). Murine mammary carcinoma 4T1 (Cat# CRL-2539) and 4T1/luc cells (Cat# CRL-2539-LUC2) and murine colon cancer CT26 (Cat# CRL-2638) cells were purchased from American Type Culture Collection (ATCC, Manassas, VA, USA). Murine pancreatic cancer Panc02 cells were kindly provided by Dr Jun Wang (South China University of Technology). The cells were grown in Roswell Park Memorial Institute (RPMI) 1640 Medium (Gibco, USA) containing 10% (v/v) fetal bovine serum (FBS, Excell Bio, China). Standard culture condition (37 °C, 5% $CO_2$) was used, and the cells were passaged when they reached 80-90% confluence. In the in vitro study, the HMVECs were seeded at initial seeding density of 40,000 cells/cm² and were cultured for another 48 h to form the monolayer endothelial cells barrier.

### Preparation of Au NPs suspension and the in vitro cell exposure
Au NPs with different surface roughness was synthesized following the previously reported protocol[12]. Briefly, spherical citrate capped Au NPs, synthesized following the established Turkevich–Frens method[31], were used as seed and the surface roughness was achieved through 4-(2-hydroxyethyl)-1-piperazineethanesulfonic acid (HEPES, Sigma Aldrich) mediated branch growth (See Supporting Information). The shape and the primary size of the Au NPs was determined through the field emission transmission electron microscope (FE-TEM, JEOL 2100-F). Briefly the Au NPs were dropped on carbon coated TEM grid and viewed under accelerating voltage of 200 kV. The Au NPs primary size was determined by measuring at least 50 randomly selected Au NPs with ImageJ software[32]. The Au NPs treatment

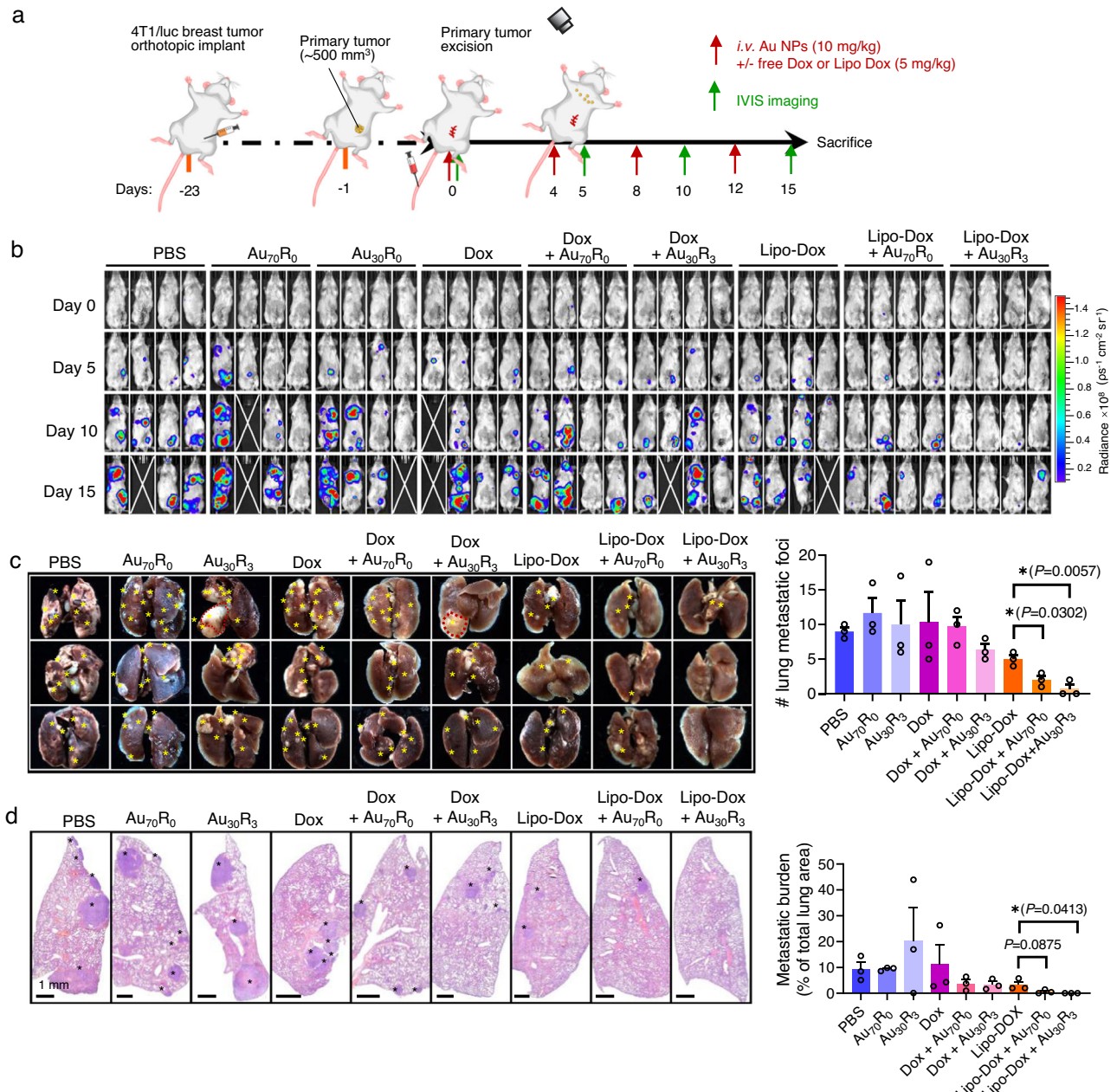

**Fig. 8 | NanoEL particles can kill micrometastases through increasing leakiness.**
**a** Mice treatment scheme. 4T1/luc breast tumor was orthotopically implanted and excised from the primary site when reaching 500 mm³. Metastatic burden was visualized with IVIS imaging system. NanoEL particles: $Au_{30}R_3$ and $Au_{70}R_0$. $n = 4$ mice/group. Treatment was carried out on Days 0, 4, 8, and 12. While IVIS imaging was conducted on Days 0, 5, 10, and 15. **b** Live animal imaging ($n = 4$/group) show a significant reduction in 4T1 metastasis found in groups receiving co-treatment of

Lipo-DOX and NanoEL Au NPs. In good agreement, **c** bright field image of the lungs ($n = 3$ mice/group) harvested from the animal and **d** H&E staining of lung sections ($n = 3$ mice/group) show a significant reduction in the number of metastatic burdens in the Lipo-Dox and NanoEL Au NPs cotreated groups. Data are mean ± SD, $n = 3$ mice/group, One-way ANOVA, Tukey HSD post-hoc test, $P < 0.05$, *significant against Lipo-Dox group. Source data are provided as a Source Data file.

suspension was maintained at the same particle number ($1 \times 10^9$, $3 \times 10^9$, and $10 \times 10^9$ particles/mL) was freshly prepared by centrifuging the necessary amount of Au NPs and re-dispersing them back in the complete EndoGRO medium with the help of sonication in the ice bath for 1 min. The HMVECs were exposed by replacing their growth medium with the Au NPs treatment suspension. Fresh growth medium was used to treat the control group. The effect of Au NPs on HMVECs' health was determined through a series of tests comprising of cell viability, membrane damage profile, and oxidative stress profile (See Supporting Information). Additionally, internalization of the Au NPs was determined with inductively coupled plasma mass

spectrophotometry (ICP-MS; See Supporting Information) with taking consideration of their sedimentation, diffusion and effective delivered dose as previously described[33–35].

For the study utilizing inhibitors, the HMVECs were pre-treated with the said inhibitors 1 h prior the Au NPs treatment. Afterward, the pre-treatment medium was fully replaced with Au NPs treatment medium supplemented with the said inhibitor. Fresh EndoGRO-MV-VEGF medium supplemented with the inhibitor was added as a control. Endocytosis inhibitors, methyl β-cyclodextrin (MβCD, 5 mM; Sigma Aldrich, USA), monodansyl cadaverin (MDC, 10 μM; Sigma Aldrich, USA), and their cocktail formulation (5 mM MβCD and 10 μM MDC)

were used to block the Au NPs internalization. In addition, Src-family protein tyrosine kinase inhibitor, PP1 (10 μM; Sigma Aldrich, USA), and Rho associated kinase inhibitor, Y27632 (10 μM; Sigma Aldrich, USA), were also used in this study.

For the study utilizing antioxidants, the cells were pre-exposed to these antioxidants for 1 h prior Au NPs treatment. Afterward, the pre-treatment medium was fully replaced with Au NPs treatment medium supplemented with the said antioxidants. Fresh medium supplemented with the antioxidants was added as a control. Antioxidants, L-reduced glutathione (GSH, 10 mM; Sigma Aldrich) and N-acetyl cysteine (NAC, 10 mM; Sigma Aldrich) were used to alleviate the ROS production.

### Transwell insert assay

The monolayer HMVECs were cultured on Transwell® insert (poly-carbonate filter, 0.4 μm pore, 24 well format; Corning Costar, USA). Following the formation of monolayer HMVECs, the growth medium was carefully removed from the insert, and the cells were exposed to different size groups and different surface roughness groups of Au NPs for 15, 30, and 60 min. Fresh cell culture medium were introduced to the control group. To detect the monolayer permeability, the treatment suspensions were supplemented with FITC–dextran (1 mg/mL, 40 kDa; Sigma Aldrich). Following the treatment, the supernatant at basolateral compartment was collected, and the FITC–dextran fluorescence signal was quantified with a microplate reader (Hidex, Finland) at excitation/emission wavelengths of 490/520 nm. The NanoEL index was determined by normalizing the fluorescent signal from the treatment group to the control group.

### Immunofluorescence

Monolayer HMVECs barrier formed on 8-well chamber slide (LabTekII, Thermofisher, USA) were exposed to Au NPs at different time (15, 30, and 60 min). Following the Au NPs exposure, the cells were fixed with the incubation of 4% paraformaldehyde (PFA; Sigma Aldrich) for 15 min. The cells then were incubated with 0.2% Triton X-100 solution (Sigma Aldrich) for 15 min to facilitate their cell membrane permeabilization. A blocking solution comprising of 2% bovine serum albumin (BSA; Sigma Aldrich) and 0.1% Triton X-100 were added afterward, and the cells were incubated for 1 h. Following the blocking step, the cells were probed with primary VE-cadherin antibody (Clone D87F2, Cell Signaling Technology, #2500, dilution 1:100) solution overnight at 4 °C, followed with Alexa 647-conjugated secondary antibody (Life Technologies, #A21443, dilution 1:400) incubation for 1 h at room temperature. To facilitate the visualization of actin filaments and nucleus, secondary antibody solution was supplemented with Alexa-Fluor 488-phalloidin stain (Life Technologies, #A12379, dilution 1:200) and Hoescht 33342 (1 μg/mL; Life Technologies), respectively. Thereafter, the cells were washed and mounted with Prolong anti-fade mounting medium (Life Technologies). Immunofluorescence images were obtained with inverted Leica Epifluorescence microscope (Leica DMI 6000, Germany) utilizing Fluotar 63x/1.25 oil immersion microscope objective. Phosphate buffered saline (PBS, pH 7.4) was used for general washing and solutions preparation. Antibody solutions used in this study were prepared in 0.2% BSA, 0.1% Triton X-100 solution. The number of NanoEL gaps and the gaps size were analyzed with ImageJ software[32].

### Intravital microscopy imaging

The following methods were utilized for studies with ectopic tumor models. Cell suspension (4T1 cells or CT26 cells, $5 \times 10^5$ cells in $1 \times$ PBS) were subcutaneously implanted into one of the ears of the female BALB/c mice. The ectopical pancreatic cancer model was developed by injection of $1 \times 10^6$ Panc02 cells into one of the ears of the male C57BL/6 mice. When the tumor volumes were around 50 mm$^3$, mice (n = 3 for each group) received intravenous injection of vehicle control, or

$Au_{30}R_3$ NPs (10 mg/kg, containing 1% BSA in PBS). At 2.5 h after injection of the above formulations, mice were injected intravenously with 3,3′-Dioctadecyloxacarbocyanine perchlorate (DiO)-labeled polymeric NPs (defined as NPs-DiO), which was self-assembled by poly (ethylene glycol)-block-poly (D, L-lactide) ($PEG_{5k}$-b-$PLGA_{11k}$) at 0.25 mg/kg of DiO. An inverted laser scanning microscope (Nikon Ti2, Japan) with a 10× air 0.45 NA or 20× water immersion 0.95 NA objective lens was used for intravital microscopy. Software NIS-Elements AR (v. 5.20.00) was used for image acquisition. During imaging, mice were sedated with isoflurane inhalation anesthesia 1.5 ~ 2 % (v/v) isoflurane in $O_2$ and placed within a custom-designed imaging box. Mice were maintained under anesthesia on a heating pad kept at 37 °C. For the study of PP1 inhibition in vivo, 4T1-bearing mice were intravenously injected with PP1 (1.5 mg/kg, Selleck, USA) 1 h before the Au NPs treatment.

The following methods were utilized for studies with orthotopic tumor models. Orthotopic mammary tumor models were prepared by injecting $5 \times 10^5$ 4T1 cells (50 μL) with 20% Matrigel (BD Bioscience, USA) into the mammary fat pad of female BALB/c mice (6–8 weeks). Orthotropic pancreatic tumour models were developed by injecting $1 \times 10^6$ Panc02 cells into the pancreas of male C57BL/6 mice (6–8 weeks). The tumor was allowed to grow to 80–100 mm$^3$ over 1–2 weeks. Mice (n = 3 for each group) were intravenously injected with vehicle control (1% BSA in PBS), or $Au_{30}R_3$ NPs (10 mg/kg, containing 1% BSA in PBS). At 2.5 h after injection of the above formulations, mice were intravenously injected with NPs-DiO. The tumor was carefully separated from underlying tissues, and the mouse was placed on the heated stage of the inverted confocal microscope.

### In vivo distribution and ex vivo tumor accumulation

Murine breast cells 4T1 ($5 \times 10^5$) were ectopically or orthotopically implanted on female BALB/c mice (n = 4 or 5 for each group; 6–8 weeks). Mice were intravenously injected with PBS (containing 1% BSA), $Au_{70}R_0$ NPs (10 mg/kg, containing 1% BSA) or $Au_{30}R_3$ NPs (10 mg/kg, containing 1% BSA). At 2.5 h after injection, mice were intravenously injected with 1,1′-Dioctadecyl-3,3,3′,3′-tetramethylindodicarbocyanine perchlorate (DiD)-labeled polymeric NPs (NPs-DiD). The dose of DiD was 0.25 mg/kg. At the preset times, in vivo fluorescence images were acquired on the Xenogen in vivo imaging system (IVIS) Lumina system (Caliper Life Sciences, USA) using Living Image software (v. 4.5). Moreover, after 24 h post-injection, the organs of mice including heart, liver, spleen, lung, kidney, and tumor were harvested. Then fluorescence images were also acquired on the Xenogen IVIS Lumina system as well. The tumor tissues were collected, embedded in OCT compound, and sectioned at 1 mm (top plane), 3 mm (middle plane), and 5 mm (bottom plane), respectively. Cell nuclei were stained with DAPI. Images of the slide were acquired with the automated quantitative microscopy-based image analysis system TissueFAXS PLUS (TissueGnostics GmbH, Austria; software v. 4.2) with the X20 objective.

### Histology and immunohistochemistry

Female BALB/c mice (6–8 weeks) bearing 4T1 tumors were intravenously injected with vehicle control (1% BSA in PBS), $Au_{70}R_0$ NPs (10 mg/kg, containing 1% BSA in PBS) or $Au_{30}R_3$ NPs (10 mg/kg, containing 1% BSA in PBS). At 4 h post-injection, the Au NPs treatment tumor and control tumor were harvested. The tumor tissues were fixed in 4% paraformaldehyde and embedded in paraffin. Subsequently, the tumor tissues were sectioned and stained with hematoxylin and eosin (H&E).

For the immunohistochemical studies, tumor tissues were harvested, fixed with 4% PFA, embedded in paraffin, and then sectioned into 5 μm section. Subsequently, the sections were dewaxed and rehydrated with xylene, ethanol, and deionized water. The slides were boiled for 90 s in 0.01 M sodium citrate buffer (pH 6.0) in the pressure cooker for antigen retrieval, followed by blocking with 10% normal goat serum. The slides were stained with the CD31 antibody (Clone

D8V9E, Cell Signaling Technology, #77699, dilution 1:100), VE-Cadherin antibody (Abcam, #ab33168, dilution 1:50) or phospho-VE-cadherin (Tyr658) antibody (Thermo Fisher Scientific, #44-1144 G, dilution 1:50) overnight at 4 °C, and then reacted with horseradish peroxidase-conjugated secondary antibody for 1 h at room temperature.

## Biosafety analysis

Female BALB/c mice ($n = 5$ for each group; 6–8 weeks) were intravenously injected with vehicle control (1% BSA in PBS), $Au_{70}R_0$ NPs (10 mg/kg, containing 1% BSA in PBS) or $Au_{30}R_3$ NPs (10 mg/kg, containing 1% BSA in PBS). At 24 h after injection, mice were sacrificed, and the hearts, livers, kidneys, lungs, and spleens were harvested and fixed in 4% PFA for H&E staining. Blood cell numbers were counted, and serum alanine aminotransferase (ALT), aspartate aminotransferase (AST), and creatinine (CRE) concentrations were measured by a commensal kit (Nanjing Jiancheng Bioengineering Institute, China).

## Antitumor study

For antitumor study with early-stage tumors, female BALB/c mice (6–8 weeks) were divided into different groups randomly ($n = 7$ for each group). 4T1-luc cells ($5 \times 10^5$) suspended in PBS were subcutaneously injected into the left flank of each mouse. When the tumor volumes were around 80 mm³, the mice were intravenously injected with various 10 mg/kg Au NPs formulations ($n = 7$ for each group). At 2.5 h after injection, the mice were injected intravenously with Doxorubicin (Dox, Meilunbio, China), or liposomal Doxorubicin (Lipo-Dox, Shanghai Fudan-Zhangjiang Bio-Pharmaceutical Co., Ltd, China) at an equivalent dose of 5 mg/kg Dox. The following nine formulations were used in this study: (i) PBS, (ii) $Au_{70}R_0$, (iii) $Au_{30}R_3$, (iv) Dox, (v) Dox + $Au_{70}R_0$, (vi) Dox + $Au_{30}R_3$ (vii) Lipo-Dox, (viii) Lipo-Dox + $Au_{70}R_0$, and (ix) Lipo-Dox + $Au_{30}R_3$. The tumor growth was monitored by measuring the perpendicular diameter of the tumors (i.e., length and width, respectively) using calipers every two days. The estimated volume was calculated according to the formula:

$$Tumor\ volume\ (mm^3) = 0.5 \times length \times (width)^2 \quad (1)$$

Weight of each mouse was also measured every two days. At the end of the antitumor study, the mice were sacrificed right after 10 min being intraperitoneal injected with the relevant substrate, and the organs (e.g., liver, lung, heart, spleen, and kidney) harvested for imaging with bioluminescence imaging system, which was carried out using an IVIS spectrum system.

For antitumor study with late-stage tumors, female BALB/c mice (6–8 weeks) mice were divided into different groups randomly ($n = 6$ for each group). 4T1-luc cells ($5 \times 10^5$) suspended in PBS were subcutaneously injected into the left flank of each mouse. When the tumor volumes were around 500 mm³ (25 days post inoculation), the mice were intravenously injected with various 10 mg/kg Au NPs formulations ($n = 6$ for each group). At 2.5 h after injection, the mice were injected intravenously with liposomal Doxorubicin (Lipo-Dox) at an equivalent dose of 10 mg/kg Dox. The following nine formulations were used in this study: (i) PBS, (ii) $Au_{70}R_0$, (iii) $Au_{30}R_3$, (iv) Lipo-Dox, (v) Lipo-Dox + $Au_{70}R_0$, and (vi) Lipo-Dox + $Au_{30}R_3$. Treatments were only carried out on Day 0 and Day 6. The tumor growth was monitored by measuring the perpendicular diameter of the tumors (i.e., length and width, respectively) using calipers on days 2, 4, 6, 8, 10, 11, and 12 following Eq. 1. The weight of each mouse was also measured every two days. At the end of the experiment (Day 12), IVIS Spectrum imaging system was utilized to visualize the tumors by an intraperitoneal injection of D-luciferin, potassium salt (75 mg/kg, Gold Biotechnology). Imaging was done for all animals in the group ($n = 6$ mice /group) except for PBS vehicle control group ($n = 5$ mice/group) of which 1

animal died prior to the live imaging analysis. At the end of the antitumor study, the mice were sacrificed, and the tumor was harvested.

## Metastasis therapy

Female BALB/c mice (6–8 weeks) were inoculated with 4T1/luc cells ($5 \times 10^5$ cells per mouse) via an injection into the mammary fat pad. Tumors were resected from the primary site when reaching ~500 mm³. Immediately after surgery, the mice were intravenously administered with various 10 mg/kg Au NPs formulations ($n = 4$ for each group). At 2.5 h after injection, the mice were intravenously injected with Dox, or Lipo-Dox at an equivalent dose of 5 mg/kg Dox every 4 days. Mice were administered with substrate before bioluminescence imaging using an IVIS spectrum system. At the end of therapy (day 15), the mice's lungs were analyzed ex vivo and H&E staining was conducted.

## Statistical analysis

Each experiment was tested with a minimum of three biological replicates. Data reported are mean ± standard deviation (SD) or mean ± standard error of mean (SEM). Statistical analysis was conducted with Origin 9.5 (Originlab, USA) or Prism 8 (GraphPad, USA), and its statistical significance was ascertained when $P < 0.05$. One-way ANOVA was used to make statistical comparison for studies with more than 2 groups, while Two-way ANOVA repeated measures was employed for time course studies.

## Reporting summary

Further information on research design is available in the Nature Portfolio Reporting Summary linked to this article.

## Data availability

Data are available within the Article, Supplementary Information or Source Data file. Source data are provided with this paper.

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

## Acknowledgements

This study was supported by Singapore Ministry of Education Academic Research grants, R-279–000-418–112 (DTL), R-279-000-498-114 (DTL), R148–000–217–112 (DTL); National Key R&D Program of China: 2020YFA0710700 (YW) and 2022YFC2409700 (PCK); National Natural Science Foundation of China grants: 52025036 (YW) and T2250710182 (PCK).

## Author contributions

Conceptualization: D.T.L.; Methodology: D.T.L., M.I.S., Q.W., Y.W.; Investigation: M.I.S., Q.W., N.Y., and J.T.; Data analysis: D.T.L., M.I.S., Q.W., and Y.W.; Funding acquisition: D.T.L., Y.W.; Supervision: D.T.L., Y.W.; Writing – original draft: D.T.L., M.I.S., Q.W., and Y.W.; Writing – review & editing: D.T.L., M.I.S., Q.W., Y.W., H.K.H., K.A., and P.C.K.

## Competing interests

The authors declare no competing interests.
