## [Peer review file · Nature Communications]

REVIEWER COMMENTS

Reviewer #1 (Remarks to the Author):

In this paper, the authors described a strategy to enhance the therapeutic effect of nanomedicine by increasing the tumor vascular leakiness using Au NPs that interacts and disrupts adherens junction protein cadherin. The idea is very interesting and of clinical importance, and the authors provided sufficient results to support the conclusions. I think this paper is suitable and acceptable for publication in Nature Communications. The followings are my detailed comments which may be needed to be considered and clarified, for better understanding this story.

1. The authors also addressed that this strategy is a double-edge sword, so it is important to control and make sure the effect of NanoEL is temporary, so it is helpful to show some results and more discussions regarding the optimal dosing and timing of NanoEL.
2. L337-338, the authors mentioned that “micrometastases could not stimulate neo-vascularization and thus not therapeutic leaky”, it is not completely true, actually some papers clearly showed the abnormal and leaky blood vessels in metastatic tumor nodules and accumulation of nanodrugs also found in metastatic tumors (10.1111/j.1349-7006.2010.01619.x, 10.1016/j.ejpb.2018.06.005.).
3. The authors used different in vivo solid tumor models, and the treatment was carried out at the early stage of tumor, however, considering that most of clinical tumors for chemotherapy are late stage progressive tumors, it may be important to challenge this strategy in tumors of later stages.
4. It is still not very clear that why NanoEL did not affect normal blood vessels, because in vitro study using normal vascular cells showed clearly the effect of enlarging the gaps between cells. I supposed this should be at least partly due to the EPR effect, namely NanoEL accumulates in tumor site by EPR effect, then fulfill its effect mostly in tumor tissue, that means the EPR effect-based accumulation of NanoEL is the first basic step to achieve its effect. In this context, the EPR effect should be addressed in more details and discussions regarding this issue are preferred.
5. L91, 103, Figure 1C does fit to the descriptions of the size changes of NanoEL.

Reviewer #2 (Remarks to the Author):

In this study, authors have characterized new Au NPs that vary in their size, surface roughnesses and promote endothelial leakiness (EL) in a roughness dependent manner. This effect of rough Au NPs was found to involve, as was characterized in their previous works involving other NP formulations - Src dependent phosphorylation of VE-Cadherin leading to Wnt dissociation, loss of VE-Cadherin via

internalization and downstream activation of Rho/ROCK dependent actomyosin contractility leading to reduction in cell-cell contact area. Using ectopic tumor models involving intravital imaging, authors report Nano EL to occur specifically within tumor vasculature in mice treated with rough Au NPs. They also validate their findings in orthopic models and find increased retention of conjugated NP-DiD within the tumors of mice specifically treated with rough Au NPs, effects that reversed with the treatment of Src inhibitor. Interestingly, authors report that treatment with rough Au NPs resulted in increased gap formation between endothelial cells as well as enlarged the blood vessels in vivo and that too specifically within tumor vasculature. Finally, authors show that co-treatment of tumor-bearing mice specifically with Doxorubicin and rough Au NPs combination significantly reduces tumor burden at the primary site as well as reduces micrometastases at distant sites, suggesting Au NP treatment facilitated increased access of Dox to tumor sites.

The use of - in vitro and in vivo models, intravital and whole-body imaging, multiple assays – are impressive and provide valuable information. However, there are a few concerns/gaps in the approach, analyses and interpretation of their results that need to be addressed to truly distinguish this study from their previous works and highlight the advancement in the Nano EL approaches the authors are reporting in this study.

Major concerns

1) The key finding of this study is - smaller size in combination with increased roughness of Au NPs induces robust NanoEL and can aid tumor-targeting in vivo. This aspect must reflect in their title, abstract and discussion, which is currently lacking as the current text portions appear redundant to their previous works and robs the focus away from the new results they present in this study.

2) Why does increase in surface roughness of Au NPs lead to enhanced Src-dependent VEC phosphorylation and associated downstream signaling? What is the underlying mechanism behind this? Is it increased shear stress or a direct interaction of Au NPs with specific portions of the endothelial plasma membrane? This aspect needs to be addressed to put their findings in perspective and differentiate this work from their past NP formulations.

3) Based Fig. S7A, larger NPs (Au70) get internalized greater than the smaller counterparts (Au30).

a) Is the reduced activity of Au70 NPs, when compared to Au30 NPs, due to their enhanced internalization? Clearly MDC and MbCD cocktail fails to completely block internalization of NPs (statistical significance doesn't mean internalization was totally blocked!).

b) Given the nature and sizes of the particles, it is probable that NPs internalize via fluid phase uptake/micropinocytosis and not via Clathrin/Caveolin pathway. Have the authors tried EIPA or Latrunculin treatments to block fluid phase uptake and assess Au NP internalization and score for signaling/EL effects?

4) Authors' previous work (2013 Nat. Comm. paper) has shown that NanoEL can also be induced in the subcutaneous vessels in mice, if introduced locally.

a) What is so distinct about tumor endothelial cells, and not those in other tissue vessels, that makes it highly responsive to rough Au NP-induced NanoEL in this study? It is difficult to imagine that rough Au NPs induces NanoEL only within the tumor tissue and nowhere else/normal tissues in mice!

b) Based on results in Figs. 4 B-E, the NP-DIDs preferentially get enriched and retained within tumor more than other tissues. Clearly, tumor tissue inherently possesses higher blood flow and gaps compared to other tissues, thereby acting as a sink to NPs in general. Therefore, one could argue that rough Au NPs probably also preferentially localize to tumor vasculature due to increased blood flow, leading to exaggeration in vasodilation and thereby resulting in excessive leakage. Have the authors quantified Au NP amounts in the tumor as against lungs, liver other tissues? Perhaps that might explain the observed effects reported here.

c) A related question is – have the authors injected NP-DIDs in control animals (without tumors) and looked for leakage/retention in different tissues/whole body imaging, with and without Au NP treatment? Minor effects might not reflect in the parameters tested in Fig. S24.

d) Alternately, does Au NPs only work to exaggerate vessels already experiencing vasodilation? One can test this by locally inducing vasodilation and use intravital imaging to score for EL with and without rough Au NPs (via i.v) in an accessible vasculature-rich site such as skin flap or cremaster muscle etc.

Minor concerns

1) In page 4, lines 4 & 8, shouldn't Fig.1C be 1A?

2) It is unclear as to how many independent experiments were analyzed using TEM and represented? This needs to be reported.

3) For the analysis of gap numbers and area analyzed using IFs in Fig.1, n=5 images or ROIs per group represent how many images from how many independent experiments for each condition? Also is the data represented as mean+/-SD? Ideally, this should be represented as average number of gaps/area per independent experiment and represented as mean+/- SE.

4) In Fig. 3 D-E (& later) involving intravital imaging, n=12 images/ROIs represent how many images per experiment, 4? Unless reporting individual cell or vessel behavior, data should be reported as an average/mean per experiment (1 data point for each mouse) and reported as mean+/-SE with the statistical analysis.

5) In Fig. 3 J-K, how many mice per experiment and how many experiments does the data represent? Also, is the mean represented without an error bar, if so, why?

6) In Fig. 7 C-D, there is a lot of variability in the data points in each group, especially in the 'no Dox' control tumor condition and is probably due to just 3 mice per group experiment. Authors need to supplement this with another experiment to be able to reliably interpret and report results for this part.

Response to Reviewers' Comments (NCOMMS-22-46632.R1)

Reviewer #1 (Remarks to the Author):

In this paper, the authors described a strategy to enhance the therapeutic effect of nanomedicine by increasing the tumor vascular leakiness using Au NPs that interacts and disrupts adherens junction protein cadherin. The idea is very interesting and of clinical importance, and the authors provided sufficient results to support the conclusions. I think this paper is suitable and acceptable for publication in Nature Communications. The followings are my detailed comments which may be needed to be considered and clarified, for better understanding this story.

1. The authors also addressed that this strategy is a double-edge sword, so it is important to control and make sure the effect of NanoEL is temporary, so it is helpful to show some results and more discussions regarding the optimal dosing and timing of NanoEL.

Response: We thank the reviewer for this critical comment. Indeed, in our additional experiment we showcase that optimal timing of the NanoEL determine the therapeutic success of the nanomedicine. a permanent or temporal phenomenon. Utilizing IVM analysis on ectopic 4T1 breast cancer model over the course of 10 hours we observed that the NanoEL induction to be temporal in nature. Leakiness, evidenced by the presence of DiO-NPs tracers in the tumor interstitial space, was noted as early as 70 min post tracer injection and persisted at least for another 440 min (~ more than 7h) before dissipating back to the basal level 540 min post tracer injection. We previously demonstrated in in vitro endothelial monolayer model that TiO₂ NPs were able to induce similar leakiness for at least 6 h (Peng et al., Nature Nano 2019). In contrast, thrombin, a well-known inducer of vascular permeability is only effective opens vasculature for 2.5h. This showcases the therapeutic window which could utilize to facilitate the nanomedicine delivery. This new additional data is presented in **Figure S16**.

Figure S16. Transient induction of NanoEL in 4T1 tumor model. Murine 4T1 breast tumor cells were ectopically implanted on the mice (n=3 mice/group). Thereafter, the mice were treated either with vehicle control or NanoEL particles, Au₃₀R₃ (10 mg/kg). NPs-DiO (~100 nm) were used as common visualization particles, and the in vivo leakiness was viewed through

intravital imaging (IVM) over the course of 10 hours. **(A)** Representative multicolor images taken over the course of 10 hours show increase leakiness in Au₃₀R₃ nanoparticles group is transient in nature. Scale bar: 50 μm. **(B)** Corresponding quantification of the efflux intensity in the interstitial area gave evidenced on the temporal effect of Au₃₀R₃ induced leakiness. Data are mean ± SEM, n = 3 mice/group.

2. L337-338, the authors mentioned that “micrometastases could not stimulate neo-vascularization and thus not therapeutic leaky” it is not completely true, actually some papers clearly showed the abnormal and leaky blood vessels in metastatic tumor nodules and accumulation of nanodrugs also found in metastatic tumors (10.1111/j.1349-7006.2010.01619.x, 10.1016/j.ejpb.2018.06.005.).

Response: Thank you for this insightful comment and suggestion of the publications. Prof Maeda was the intellectual driving force that motivated our work into the interactions between nanomaterials and the endothelium. The two highlighted papers are well known in the field.

Pertaining the comment on metastatic tumors, we would like to refer to Figure 6 of the first publication mentioned by the reviewer (Daruwalla et al. Cancer Science 2010; screenshot below)

Fig. 6. Livers at day 21 (72 h after styrene-maleic acid copolymer (SMA)-pirarubicin injection) were excised and fixed in formalin for histological analysis. The percentage of tumor necrosis was quantified on H&E-stained paraffin sections. (a) SMA-pirarubicin (100 mg/kg) resulted in 3-fold increase in tumor necrosis (***P* < 0.001). (b) Control tumors demonstrated poor-to-moderate differentiation. Tumors were cohesive (T) with minimal central necrosis and displaced the surrounding normal liver (NL). (c) Drug treated tumors were extensively damaged; high degree of central necrosis extending to the periphery (black arrows). (c–e) Treated tumors were surrounded by a fibrous capsule which was absent in untreated tumors. (d) Insert, fibrous capsule. Treated tumor cells exhibited hydroptic swelling and enlarged nuclei. (f) Scanning electron micrograph of untreated tumor demonstrating dilated tumor vessels (TV). (g) Following treatment, the central tumor core is void of vessels, and appears necrotic (N). Vessels from the periphery have become occluded resulting in tapered ends (white arrows) indicative of vascular shut down – a potential mechanism of tumor destruction. Scale bars, (a–c) 200 μm, (d) 100 μm.

The focus of this Cancer Science paper is in the highly efficacious outcomes of their polymeric micelles pirarubicin drug delivery systems in addressing both a flank implanted tumor and a liver metastasized tumor and not to study the vasculature of a metastatic tumor. The liver metastasis model was created via an intrasplenic injection of a large number of cells (50,000 MoCR) which then following the corresponding veins into the liver to form the metastatic tumor-in-liver model. The tumor cells after 16 days of in vivo hepatic growth certainly grew to be so large that it occluded part of the host liver tissues (see above screenshot of Figure 6 from the Daruwalla et al. Cancer Science 2010). Based on Figure 6b of the control tumor (no

treatment rendered), the tumor diameter is approximately 1000um or 1mm in diameter. At 1mm diameter size, the tumor would be able to secrete angiogenic factors (Carmeliet and Jain, Nature 2000; Fukumura and Jain, Journal of Cellular Biochemistry 2007; Folkman and Klagsbrun, Science 1987) to attract neo-vascularization which tend to be leaky as described by the reviewer and in Daruwalla et al. Cancer Science paper.

But this 1mm tumor far exceeds the few microns diameter of micrometastases to which we were referring. In clinical metastasis, for example in breast cancer, detection of just a few cancer cells in the axillary lymph node will be sufficient to give a metastatic prognosis that would warrant mastectomy, radiotherapy or chemotherapy. At the beginning growth stages of these few metastasized tumor cells at the secondary tumor site, the tumor is going to be very small as it only consists of a few cells; thus known to be micrometastases. At that small size, there is still no driving impetus to ramp up neo-vascularization, thus peripheral vasculature surrounding the micrometastases at the secondary site would be sufficient to deliver and meet the micrometastases' metabolic demands. But all metastasized tumors must start off small before they become big. Logically, since it is much easier to kill a small, metastasized tumor that has recently embedded itself at the secondary site than to kill a large metastasized established tumor and therefore we should attack the metastasized tumors when they are still small and unestablished and not wait until they are too large to be killed. But the issue with micrometastases is not a matter of killing efficacy of the chemotherapy but an issue of the accessibility. That is where our technology come in, by inducing the leakiness when there was none; when the metastasized tumor was still a small few-cells size.

In our original submission and also in revised **Figures 7, 8 and S34**, we upped the ante by first allowing the flank tumor to form metastases, i.e., tumor cells to metastasize out and travel to the lung and then forming a small metastasized tumor and NOT injecting large numbers in the cells into the lung to form secondary colonies such was performed by Daruwalla et al. Cancer Science 2010 (they used 50,000 cancer cells). In order to also better recapitulate the clinical practice, we surgically resected out the primary tumor after the metastasis has already occurred while the animal was still alive just like in the human patient case. As 4T1-EGFP tumor cells have been known to metastasize to the lung, we specifically zoom into the lung to look for these metastasis colonies. We search intravitaly the mice lungs to find the 4T1-EGFP micrometastases (**Figure S34A**). Utilizing CD31 counterstaining to visualize the vasculature, it could be observed there was little vasculature infiltrating within the micrometastases colony. Considering this, accessibility of the nanodrug arising from engineered leakiness out of the sparse vasculatures around the micrometastases tumor becomes even more critical which we have shown in **Figure S34B and C**.

Figure S34. NanoEL induction in micrometastasis 4T1 tumor. Mice ($n=3$ mice/group) were orthotopically implanted with 4T1-EGFP tumor of which it was resected from the primary site when reaching $\sim 500 \text{ mm}^3$. Immediately thereafter the mice received intravenous injection either with vehicle control or NanoEL particles, Au_{30}R_3 (10 mg/kg). NPs-DiD tracer and PE-conjugated anti-CD31 were used to visualize the *in vivo* leakiness and tumor neovascularization, respectively. **(A)** Representative IVM images of the lung metastatic secondary tumors. Neovascularization (grey) was not observed on the micrometastasis tumor. 4T1 tumor cells (green). Scale bar: 50 μm . **(B)** Representative IVM image and **(C)** quantification of the NPs-DiD tracer intensity show increased leakiness into the interstitial space surrounding micrometastasis in the Au_{30}R_3 nanoparticles group. Scale bar: 20 μm . Data are mean \pm SD ($n=9$ ROI from 3 mice/group).

Additional reference:

- Jurstine Daruwalla, Mehrdad Nikfarjam, Khaled Greish, Cathy Malcontenti-Wilson, Vijayaragavan Muralidharan, Chris Christophi, Hiroshi Maeda. In vitro and in vivo evaluation of tumor targeting styrene-maleic acid copolymer-pirarubicin micelles: Survival improvement and inhibition of liver metastases. *Cancer Science* 2010, 101, 1866-1874
- Peter Carmeliet and Rakesh K. Jain. Angiogenesis in cancer and other diseases. *Nature* 2000, 407, 249–257.

- Dai Fukumura and Rakesh K. Jain. Tumor microenvironment abnormalities: Causes, consequences, and strategies to normalize. *Journal of Cellular Biochemistry* 2007, 101, 937-949.
- Judah Folkman and Michael Klagsbrun. Angiogenic Factors. *Science* 1987, 235, 442-447.

3. The authors used different in vivo solid tumor models, and the treatment was carried out at the early stage of tumor, however, considering that most of clinical tumors for chemotherapy are late stage progressive tumors, it may be important to challenge this strategy in tumors of later stages.

Response: We thank the reviewer for the suggestion. We have added a new experiment of which the Au NPs efficacy to facilitate the nanomedicine delivery in late-stage tumors was investigated. In this study, we used a large solid tumor model (of $\sim 500\text{mm}^3$) and once again investigate effect of adjusting the in vivo vascular leakiness on the efficacy of liposomal Dox (Lipo-Dox) therapy. The 4T1/luc large tumors were treated with Au₃₀R₃ or Au₇₀R₀ and Lipo-Dox formulation for only two time points (Day 0 and Day 6 of the treatment regime) and the tumor size measurements (Figure 7B) and mice weight (Figure S29B) were recorded until Day 12 when the animals were sacrificed. Similar to the observation on the early-stage tumor model (Figure 6), in the absence of Lipo-Dox formulation, Au NPs groups did not show final regression of the tumor compared to no-treatment control (Figure 7B - D). Lipo-Dox only formulation was demonstrated to once again significantly slowed down the growth of the late-stage tumor (Figure 7B); whereas its co-treatment with NanoEL Au₇₀R₀ NPs was noted to stay the growth of this late stage tumor. More interestingly, increasing the vascular escape with NanoEL Au₃₀R₃ NPs was demonstrated to effectively lead to the late tumor regression (Figure 7B). This further supports that combined effects of increased NanoEL and Lipo-Dox formulation was sufficient in reducing tumor sizes (Figure 7E-F), further highlighting that increased vascular leakiness with our nanotechnology can greatly potentiate the cancer therapeutics' effect even introduced at later stage of the tumor progression. This new additional data is presented in **Figure 7**.

Figure 7. NanoEL particles improves nano drug carrier access and therapeutic outcomes in late-stage tumor. (A) Mice treatment scheme. NanoEL particles: Au₃₀R₃ and Au₇₀R₀. n=6 mice/group. Treatment was initiated when the tumor reached the volume of ~500 mm³, and only carried out on Days 0 and 6. (B) Tumor size profile throughout the entire 12 days of treatment. Au NPs treatment appeared to improve liposomal Dox delivery groups. Data are mean ± SEM, n = 6 mice/group, Two-way repeated measures ANOVA, Tukey HSD post-hoc test, *significant against Lipo-Dox group, *P* < 0.05. After 12 days, the animals were injected with luciferin and the 4T1/luc tumor was visualized using the IVIS system before being sacrificed, and the whole tumors were harvested and size-sorted (C) Representative IVIS images and (D) Quantification of total flux show reduce tumor size in the animal receiving liposomal Dox and Au NPs group. Data are mean ± SEM, n = 6. (E) Tumor images and (F) quantification data show the Lipo-Dox alone treatment showed significant inhibition of the late-stage tumor growth, but most significant regression is found those with NanoEL Au NPs co-treatment. Data are mean ± SEM, n = 6. One-way repeated measures ANOVA, Tukey HSD post-hoc test, *significant against Lipo-Dox group, *P* < 0.05.

4. It is still not very clear that why NanoEL did not affect normal blood vessels, because in vitro study using normal vascular cells showed clearly the effect of enlarging the gaps between cells. I supposed this should be at least partly due to the EPR effect, namely NanoEL accumulates in tumor site by EPR effect, then fulfill its effect mostly in tumor tissue, that means the EPR effect-based accumulation of NanoEL is the first basic step to achieve its effect. In this context, the EPR effect should be addressed in more details and discussions regarding this issue are preferred.

Response: It is still a controversy whether there is significant leakiness arising solely from the EPR. Prof Warren Chan published a controversy paper (Sindhvani et al., Nature Materials

2020) that claimed there is no observable EPR. The EPR effect is not a nanoparticle induced effect. It is only dependent on the tumor. In other words, if there is an EPR effect, it will occur regardless of whether there are nanoparticles or not. The EPR arose due to a metabolic demand from a growing tumor which activated an angiogenic event that drawn neovascularization. These immature vasculatures tend to be leaky.

In our in vitro endothelial cells studies, there are no tumor cells at all in co-culture with the endothelial cells. Therefore, any observed leakiness in the nanoparticles compared to the no-nanoparticles group was due to the nanoparticles. Thus, it is not EPR as there are no tumor or cancer cells in the in vitro system. Thus, NanoEL is a nanoparticle effect independent from the EPR effect as purported in the field.

In the in vivo non-fenestrated capillary bed in a non-cancer region, even when we treat the NanoEL nanoparticles, we observe little or no leakiness and as reported as **Figure 3C, S11B, S27 and S28** in this paper. After comparing in vitro observations and in vivo observations, we conclude that in vivo, the normal vasculatures are intrinsically not amenable to leakiness and that is to be expected as leakiness is not physiologically helpful unless in certain instances, like immune cell extravasation. The main role of normal vasculature is to carry blood and it doesn't make sense that the 'piping' is easily leaky. Endothelial maturity in vivo takes much longer time, need blood flow during development with shear forces and with currently unknown factors (Dejana. Nature Reviews Molecular Cell Biology 2004) which we could not replicate in our humble cell culture. Since the endothelial maturity results in high resistance to leakiness, the in vitro evidence tends to be more leaky vs the in vivo evidences except for the tumor case where we observed robust leakiness. We are still in the midst of figuring things out, but we believe we are on the right track and perhaps this NanoEL topic in normal vasculature vs tumor vasculature can be discussed in another paper.

In the in vivo cancer vasculature case, in the no-nanoparticles control group, we did not observe significant leakiness at the tumor itself. Thus, we also could not observe EPR effect as purported by the community. But this paper is NOT about EPR effect nor the lack of, but this story is that even in the circumstances wherever there is no EPR effect at the tumor site, we can still engineer in the leakiness with nanotechnology through the NanoEL. We no longer need to depend solely on the tumor itself to produce the EPR for us to deliver and access the tumor.

Additional reference:

- Shrey Sindhwani, Abdullah Muhammad Syed, Jessica Ngai, Benjamin R Kingston, Laura Maiorino, Jeremy Rothschild, Presley MacMillan, Yuwei Zhang, Netra Unni Rajesh, Tran Hoang, Jamie L Y Wu, Stefan Wilhelm, Anton Zilman, Suresh Gadde, Andrew Sulaiman, Ben Ouyang, Zachary Lin, Lisheng Wang, Mikala Egeblad, Warren C W Chan. The entry of nanoparticles into solid tumours. Nature Materials 2020, 19, 566-575.
- Elisabetta Dejana. Endothelial cell-cell junctions: happy together. Nature Reviews Molecular Cell Biology 2004, 5, 261-270.

5. L91, 103, Figure 1C does fit to the descriptions of the size changes of NanoEL.

Response: We thank the reviewer for pointing out this. We have corrected the text.

Reviewer #2 (Remarks to the Author):

In this study, authors have characterized new Au NPs that vary in their size, surface roughnesses and promote endothelial leakiness (EL) in a roughness dependent manner. This effect of rough Au NPs was found to involve, as was characterized in their previous works involving other NP formulations - Src dependent phosphorylation of VE-Cadherin leading to Wnt dissociation, loss of VE-Cadherin via internalization and downstream activation of Rho/ROCK dependent actomyosin contractility leading to reduction in cell-cell contact area. Using ectopic tumor models involving intravital imaging, authors report Nano EL to occur specifically within tumor vasculature in mice treated with rough Au NPs. They also validate their findings in orthopic models and find increased retention of conjugated NP-DiD within the tumors of mice specifically treated with rough Au NPs, effects that reversed with the treatment of Src inhibitor. Interestingly, authors report that treatment with rough Au NPs resulted in increased gap formation between endothelial cells as well as enlarged the blood vessels in vivo and that too specifically within tumor vasculature. Finally, authors show that co-treatment of tumor-bearing mice specifically with Doxorubicin and rough Au NPs combination significantly reduces tumor burden at the primary site as well as reduces micrometastases at distant sites, suggesting Au NP treatment facilitated increased access of Dox to tumor sites. The use of - in vitro and in vivo models, intravital and whole-body imaging, multiple assays – are impressive and provide valuable information. However, there are a few concerns/gaps in the approach, analyses and interpretation of their results that need to be addressed to truly distinguish this study from their previous works and highlight the advancement in the Nano EL approaches the authors are reporting in this study.

Response: Thank you for the encouraging comments. Indeed, we have previously demonstrated that NanoEL could be modulated through nanoparticles' size, density, and charge. Certainly, there are a degree of similarity between our previous studies and this current manuscript as we are sharing a similar system, and thus the assays employed are similar in nature. Although, these previous works have been important as they paved the way for our team to understand the NanoEL modulation, we have yet to demonstrate the therapeutical application of NanoEL. This manuscript presented a big leap towards demonstrating NanoEL therapeutic application in which we showcased that therapeutical efficacy of nanomedicine could be tuned by the nanoparticles' physical properties.

Major concerns

1) The key finding of this study is - smaller size in combination with increased roughness of Au NPs induces robust NanoEL and can aid tumor-targeting in vivo. This aspect must reflect in their title, abstract and discussion, which is currently lacking as the current text portions appear redundant to their previous works and robs the focus away from the new results they present in this study.

Response: Thank you. Our focus in this manuscript is to highlight the therapeutic effect of the NanoEL in which the nanoparticles' roughness is merely use as a tool to showcase the therapeutic tunability against tumor endothelial cells. As to not distract the readers from the main story of this manuscript we focused on highlighting the therapeutic effect instead of the modulation effect. So we hope that the reviewer do not mind us retaining the title as it is. But certainly, we are also very open to better titles. Many thanks for your suggestions. As for the abstract and discussion part, we have made some changes which we hope are better descriptions of the new aspects of this work.

2) Why does increase in surface roughness of Au NPs lead to enhanced Src-dependent

VEC phosphorylation and associated downstream signaling? What is the underlying mechanism behind this? Is it increased shear stress or a direct interaction of Au NPs with specific portions of the endothelial plasma membrane? This aspect needs to be addressed to put their findings in perspective and differentiate this work from their past NP formulations.

Response: The heightened phosphorylation and heightened associated downstream signaling is the outcome molecular markers arising from disrupted adherens junctions of endothelial cells. The cells have not evolved a biological mechanism to recognize any aspect of a nanomaterial. So rather than see the output biological markers as a lens of the kinds of nanoparticles, we saw it simply as a measure of adherens junction disruption. Our rationale lies when we also saw the same phosphorylation and Src kinase pathway activation when we use thrombin and VEGF (Setyawati et al., Nature Communications 2013) together with TiO₂ NP treatment. Thrombin and VEGF and nanoparticles are very different inducers of the same observations. What was common amongst the three classes of phosphorylation events and Src kinase pathway induction is the disruption of the adherens junction.

The key differentiation of this work over our previous NanoEL work and other work in the literature is to emphasize on the potential to engineer leakiness in the tumor vasculature to allow better therapeutics access even in circumstances when there is no observable EPR effect.

3) Based Fig. S7A, larger NPs (Au70) get internalized greater than the smaller counterparts (Au30).

a) Is the reduced activity of Au70 NPs, when compared to Au30 NPs, due to their enhanced internalization? Clearly MDC and MbCD cocktail fails to completely block internalization of NPs (statistical significance doesn't mean internalization was totally blocked!).

Response: Thank you for your thoughts. We were unclear in our explanation. NanoEL was already observed within 30mins (**See Figure S3**), but it took at least 2h-3h for NPs endocytosis to occur (Zarska et al Bioconjugate Chemistry 2016; Sahin et al., Journal of Microencapsulation 2017; Brown et al., ACS Biomaterials Science Engineering 2020; Soudeh et al., Nanomedicine 2019; Vranic et al. Particle Fibre Toxicology 2013). Since NanoEL occurs before internalization, any difference in internalization cannot account for the difference in the degree of NanoEL.

b) Given the nature and sizes of the particles, it is probable that NPs internalize via fluid phase uptake/micropinocytosis and not via Clathrin/Caveolin pathway. Have the authors tried EIPA or Latrunculin treatments to block fluid phase uptake and assess Au NP internalization and score for signaling/EL effects?

Response: Thank you. Previous studies, utilizing EIPA and its amiloride analogue, have demonstrated NPs robust uptake via micropinocytosis pathway to happen after 3 hours (Sahin et al., Journal of Microencapsulation 2017; Brown et al., ACS Biomaterials Science Engineering 2020; Soudeh et al., Nanomedicine 2019; Vranic et al. Particle Fibre Toxicology 2013). Considering that NanoEL occurs in much shorter time frame of 30 mins, we could exclude micropinocytosis to contribute meaningfully to the NanoEL event.

Additional References:

- Monika Zarska, Filip Novotny, Filip Havel, Michal Sramek, Andrea Babelova, Oldrich Benada, Michal Novotny, Hilal Saran, Kamil Kuca, Kamil Musilek, Zuzana Hvezdova, Rastislav Dzijak, Marketa Vancurova, Katerina Krejcikova, Blanka Gabajova, Hana

Hanzlikova, Lenka Kyjacova, Jiri Bartek, Jan Proska, Zdenek Hodny. Two-Step Mechanism of Cellular Uptake of Cationic Gold Nanoparticles Modified by (16-Mercaptohexadecyl)trimethylammonium Bromide. *Bioconjugate Chemistry* 2016, 27, 2558-2574

- Adem Sahin, Digidem Yoyen-Ermis, Secil Caban-Toktas, Utku Horzum, Yesim Aktas, Patrick Couvreur, Gunes Esendagli, Yilmaz Capan. Evaluation of brain-targeted chitosan nanoparticles through blood–brainbarrier cerebral microvessel endothelial cells. *Journal of Microencapsulation* 2017, 34, 659-666.
- Tyler D. Brown, Nahal Habibi, Debra Wu, Joerg Lahann, and Samir Mitragotri. Effect of Nanoparticle Composition, Size, Shape, and Stiffness on Penetration Across the Blood–Brain Barrier, *ACS Biomaterials Science Engineering* 2020, 6, 4916–4928.
- Soudeh F. Tehrani, Florian Bernard-Patrzynski, Ina Puscas, Grégoire Leclair, Patrice Hildgen, V. Gaëlle Roullin. Length of surface PEG modulates nanocarrier transcytosis across brain vascular endothelial cells. *Nanomedicine: Nanotechnology, Biology and Medicine* 2019, 16, 185 – 194.
- Sandra Vranic, Nicole Boggetto, Vincent Contremoulins, Stéphane Mornet, Nora Reinhardt, Francelyne Marano, Armelle Baeza-Squiban & Sonja Boland. Deciphering the mechanisms of cellular uptake of engineered nanoparticles by accurate evaluation of internalization using imaging flow cytometry. *Particle Fibre Toxicology* 2013, 10, 2.

4) Authors' previous work (2013 Nat. Comm. paper) has shown that NanoEL can also be induced in the subcutaneous vessels in mice, if introduced locally.

a) What is so distinct about tumor endothelial cells, and not those in other tissue vessels, that makes it highly responsive to rough Au NP-induced NanoEL in this study?

Response: Thank you for the comment.

There are major differences between what we have earlier reported in Setyawati Nat Comms 2013 and our current in vivo model.

What we believe as the main difference that can account for the different conclusions is in the administered dose of NPs. In Setyawati Nat Comms 2013, very early in our understanding of the NanoEL mechanism, specific to the subcutaneous blood vessels model, we injected as relatively much higher local dose than what we have done in this 2023 manuscript submission. The nature of subcutaneous injection is that it forms a pocket under the skin where there is little dilution, and the NPs remain entrapped which further kept the NPs at a high concentration. This may have resulted in the leakiness of normal vasculature. This submission instead had the benefit of dilution as we introduced the nanoparticles via systemic circulation and not locally. Upon saying this, we take heed of the good advice of the reviewer. We have not done a thorough study of NanoEL on normal vasculature. But within this submission, we have done the fair controls (normal vasculature in the ear vs tumor vasculature of the opposing ear but all introduced via systemic circulation). To better phrase the observations on the normal vasculature, we have revised the text to restrict the discussion of the normal vasculature as within the control groups to reduce the interpretation that the non-leaky observation in the normal vasculature is extrapolated to all normal vasculatures. The truth is that it is hard to say that within the murine's kilometers length of normal vasculature that there is absolutely no leakiness.

b) Based on results in Figs. 4 B-E, the NP-DIDs preferentially get enriched and retained within tumor more than other tissues. Clearly, tumor tissue inherently possesses higher blood flow

and gaps compared to other tissues, thereby acting as a sink to NPs in general. Therefore, one could argue that rough Au NPs probably also preferentially localize to tumor vasculature due to increased blood flow, leading to exaggeration in vasodilation and thereby resulting in excessive leakage. Have the authors quantified Au NP amounts in the tumor as against lungs, liver other tissues? Perhaps that might explain the observed effects reported here.

Response: Thank you for the comment. There is possibility that the high pressure and dysfunctionality of the tumor vasculature could lead to the Au NPs being preferentially localize in the tumor tissue which result in higher possibility for the Au NPs to interact with the VEC in the tumor vasculature, leading to the increased leakiness. We also agree that the tumor receives more volume of blood than the surrounding tissues should therefore receive more nanoparticles. More nanoparticles should result in more leakiness if the only explanation lies with just having higher blood flow. As suggested by the reviewer, we did the Au NPs distribution analysis (normalized to tissue mass) in tumor and other major organs (heart, liver, spleen, lung, kidney) which physiologically receives even more blood than the tumor because these other major organs have much more vasculature than the tumor. And yet, the most Au NPs are accumulated in the tumor instead (**Figure S20**). While this data as a standalone is unable to prove that the Au NPs must have left the tumor vasculature through induced leakiness but taking this evidence and the rest of the data, it is pointing to a NanoEL explanation at the tumor site. Indeed, we noted that there is significant accumulation of Au NPs in tumor tissue as compared to other organs.

Figure S20. Au NPs distributions in major organs and tumors. Increased Au NPs accumulation was observed in tumors. Murine breast cells 4T1 (5×10^5 cells) were ectopically implanted on female BALB/c mice ($n = 4/\text{group}$). Mice were intravenously injected with Au₃₀R₃ NPs (10 mg/kg, containing 1% BSA). At 12 h after injection, the organs of mice including heart, liver, spleen, lung, kidney, and tumor were harvested, rinsed with PBS, weighed after removing excess fluid and homogenized. Then the accumulated Au NPs in the organs were quantified using inductively coupled plasma mass spectroscopy (ICP-MS). Data are mean \pm SEM, One-way ANOVA Tukey HSD post-hoc test, *significant against compared groups, $P < 0.05$.

c) It is difficult to imagine that rough Au NPs induces NanoEL only within the tumor tissue and nowhere else/normal tissues in mice!

Response: While we have not made such a claim of universality of non-leakiness in normal vasculature, we are glad that the reviewer surfaced this reasonable extrapolation. Please allow us to try to respond to this point through our responses to your other related queries.

A related question is – have the authors injected NP-DiDs in control animals (without tumors) and looked for leakage/retention in different tissues/whole body imaging, with and without Au NP treatment? Minor effects might not reflect in the parameters tested in Fig. S24.

Response: Thank you for the comment. We have added the experiment as the suggested by the reviewer in which control animals with no tumor were treated with Au NPs and injected with NP-DiD tracer prior to being sacrificed and the normal organs were visualized with IVIS imaging (**Figure S28A**). We selected those organs which received the highest volumes of blood presumably that if there is any robust leakiness in the normal vasculature of these organs, we should be able to find increased accumulation of the fluorescent NPs-DiD in those organs. Presumably, unusually high NanoEL related leakiness will culminate in accumulation of our tracer nanoparticles. What we found was surprising to us in that, Au NPs treatment resulted in no discernible difference in the NP-DiDs accumulation in the normal organs (**Figure S28 B-C**).

Figure S28. NanoEL is not detected in normal tissue. (A) Schematic of treatment showing normal mice bearing no tumor (n=5 mice/group) treated with 10 mg/kg Au₃₀R₃, Au₇₀R₀. BSA (1%) was used as vehicle control, and NPs-DiD (0.25 mg/kg) was used to visualize the organs. (B) Representative IVIS images and (C) Quantification of the average radiant efficiency of the organs show no discernible change in the DiD accumulation profile between the treatment groups, suggesting Au NPs did not induce leakiness in normal tissue. Data are mean ± SEM, n=5 mice/group.

Based on evidence, we cannot conclude that NanoEL has resulted in robust leakiness within the listed organs that received the majority volume of the overall blood in the animal. We are also careful in not saying that this is true for the other organs that we have not looked at as this study is still limited. This question of whether NanoEL affects normal vasculature needs some answers in future studies which is a fair point to make.

d) Alternately, does Au NPs only work to exaggerate vessels already experiencing vasodilation? One can test this by locally inducing vasodilation and use intravital imaging to score for EL with and without rough Au NPs (via i.v) in an accessible vasculature-rich site such as skin flap or cremaster muscle etc.

Response: Thank you. Vasodilation relates to the overall vasculature, while what we observed is a more localized phenomenon. We are unsure of what the kind reviewer meant with the term “exaggerate vessels”. Sorry. However, the reviewer did raise a good point. The vasodilation observation may or may not be linked to the NanoEL effect after all. If it is linked, is vasodilation part of the cause or part of the outcomes of NanoEL? That can be a topic for future exploration to be lumped together on investigating the effects of NanoEL on normal vasculature.

Minor concerns

1) In page 4, lines 4 & 8, shouldn't Fig.1C be 1A?

Response: Thank you. We have corrected the information in the text.

2) It is unclear as to how many independent experiments were analyzed using TEM and represented? This needs to be reported

Response: Three independent experiments were analyzed and represented. We have added the images and information in our manuscript (**See Figure S8**)

3) For the analysis of gap numbers and area analyzed using IFs in Fig.1, n=5 images or ROIs per group represent how many images from how many independent experiments for each condition? Also is the data represented as mean \pm -SD? Ideally, this should be represented as average number of gaps/area per independent experiment and represented as mean \pm - SE.

Response: Thank you. We have corrected the data presentation in Figure 1c. The data presented in the revised manuscript was derived from 4 ROI /group with 3 independent biological replicates. We have corrected the data presentation as mean \pm SEM.

The experiment was independently repeated for three times in which each group was tested in duplicate. Data represented was derived from one of the three independent repeats. Data represented as mean \pm SD. We have added the information in our manuscript.

4) In Fig. 3 D-E (& later) involving intravital imaging, n=12 images/ROIs represent how many images per experiment, 4? Unless reporting individual cell or vessel behavior, data should be reported as an average/mean per experiment (1 data point for each mouse) and reported as mean \pm -SE with the statistical analysis.

Response: The data was derived from 4 images/mouse and 3 mice/group. We have corrected the data presentation as mean \pm SEM.

5) In Fig. 3 J-K, how many mice per experiment and how many experiments does the data represent? Also, is the mean represented without an error bar, if so, why?

Response: Time lapse experiment which were presented in the Figure 3J-K was derived from our IVM observation of leakiness from 1 mouse/group. Hence there is no error bar for the data.

6) In Fig. 7 C-D, there is a lot of variability in the data points in each group, especially in the 'no Dox' control tumor condition and is probably due to just 3 mice per group experiment. Authors need to supplement this with another experiment to be able to reliably interpret and report results for this part.

Response: Thank you for the comment. For experiment which were originally presented in Figure 7, we started the study with 4 mice/group. However, some of the mouse (especially those which didn't receive the formulation of Lipo-DOX + Au NPs) did not survive due to the metastatic burden. At the end of the experiment, we ended up with 3-4 mice/group for further analysis. Hence, data presented in Figure 7C-D can only derived from the surviving mice/group.

REVIEWERS' COMMENTS

Reviewer #1 (Remarks to the Author):

The revised MS answered and clarified the points raised by the reviewers, I think the current version is acceptable.

Reviewer #2 (Remarks to the Author):

I am satisfied with the authors' response, revised manuscript and extensive work. I recommend its publication.

Response to Reviewers' Comments (NCOMMS-22-46632.R2)

Reviewer #1 (Remarks to the Author):

The revised MS answered and clarified the points raised by the reviewers, I think the current version is acceptable.

Reviewer #2 (Remarks to the Author):

I am satisfied with the authors' response, revised manuscript and extensive work. I recommend its publication.

Response: We thank both reviewers for their support in our manuscript publication. We also greatly appreciate their critical comments and constructive advice that help us improve the study.